# Positive-Unlabeled Learning using Random Forests via Recursive Greedy Risk Minimization

**Jonathan Wilton**[1], **Abigail M. Y. Koay**[2], **Ryan K. L. Ko**[2], **Miao Xu**[2,3], **Nan Ye**[1]

[1]School of Mathematics and Physics, The University of Queensland
[2]School of Information Technology and Electrical Engineering, The University of Queensland
[3]RIKEN, Japan 103-0027

## Abstract

The need to learn from positive and unlabeled data, or PU learning, arises in many applications and has attracted increasing interest. While random forests are known to perform well on many tasks with positive and negative data, recent PU algorithms are generally based on deep neural networks, and the potential of tree-based PU learning is under-explored. In this paper, we propose new random forest algorithms for PU-learning. Key to our approach is a new interpretation of decision tree algorithms for positive and negative data as *recursive greedy risk minimization algorithms*. We extend this perspective to the PU setting to develop new decision tree learning algorithms that directly minimizes PU-data based estimators for the expected risk. This allows us to develop an efficient PU random forest algorithm, PU extra trees. Our approach features three desirable properties: it is robust to the choice of the loss function in the sense that various loss functions lead to the same decision trees; it requires little hyperparameter tuning as compared to neural network based PU learning; it supports a feature importance that directly measures a feature's contribution to risk minimization. Our algorithms demonstrate strong performance on several datasets. Our code is available at `https://github.com/puetpaper/PUExtraTrees`.

## 1 Introduction

Positive and unlabeled learning (PU learning) has attracted increasing interest recently, due to its broad applications, including disease gene identification [24], landslide susceptibility modeling [29], inference of transcriptional regulatory network [23], cybersecurity [30] and many others [2].

Existing algorithms generally reduce PU learning to solving one or more supervised learning problems. A naive approach is to simply learn a classifier by treating all unlabeled examples as negative, but unlabeled examples can be either positive or negative. Various approaches have been developed to better handle the uncertainty on the labels of the unlabeled examples. A common approach follows an iterative two-step learning process: the first step identifies reliable negative examples, then the second step learns a model using the positive examples, the reliable negative examples and possibly unlabeled examples [22, 15]. Another common approach transforms the PU dataset into a weighted fully labeled dataset [20, 12, 11, 16]. The transformed dataset is often constructed so that the empirical risk on the transformed dataset provides a good estimate of the true risk. Depending on the assumptions on how PU data is generated, different transformations have been developed.

While random forests are known to perform well for many supervised learning problems and in principle tree-based learning methods can be applied in the above approaches, little work has been done to explore the potential of tree-based methods for PU learning. On the other hand, current state-of-the-art PU learning methods are often based on deep neural networks [16, 6]. This is partly due to the recent advances of deep learning, which allows training of powerful neural network models

with relatively little effort. In addition, while it is easy to train a neural network to minimize different loss functions, it is not clear how this can be done for tree-based methods (without overfitting the dataset), because existing tree-based methods have not been designed to directly minimize the loss. In particular, it is not clear how this can be done for nonstandard risk estimators such as the non-negative PU risk (nnPU), which has already been observed to work well with neural networks [16].

We fill this gap by providing a theoretically justified approach, which we call *recursive greedy risk minimization*, to learn random forests from PU data in this paper. Our contributions are as follows

- A new interpretation to existing decision tree learning algorithm that links the impurity reduction based learning algorithms to empirical risk minimization.
- Efficient new decision tree and random forest algorithms that directly minimize PU-data based estimators for the expected risk.
- Our approach has three desirable properties: it is robust to the choice of the loss function in the sense that various loss functions lead to the same decision trees; it requires little hyperparameter tuning as compared to neural network based PU learning; it supports a feature importance that directly measures a feature's contribution to risk minimization.
- Our algorithms demonstrate strong performance on several datasets.

We briefly review decision tree learning and PU learning in Section 2, introduce our recursive greedy risk minimization approach for decision tree learning in Section 3 and the PU version of extra trees in Section 4, present experiment results in Section 5, discuss related works in Section 6, and conclude in Section 7.

## 2    Background

**Decision tree learning from positive and negative data**    A decision tree is generally constructed by recursively partitioning the training set so that we can quickly obtain subsets which are more or less in the same class.

Algorithm 1 shows a generic decision tree learning algorithm. We start with a single node associated with the entire set of training examples. When we encounter a node $\kappa$ associated with a set $S$ of training examples, we compute the prediction value at $\kappa$ if the termination criterion is met. If not, we first compute an optimal split $(f, t)$ for the dataset $S$, where a split $(f, t)$ tests whether a feature $f$ is larger than a value $t$, and the quality of a split is usually measured by its impurity reduction. Based on

| **Algorithm 1:** LearnDT($\kappa, S$) |
| --- |
| **1** Notations: $\kappa$ - node; $S$ - dataset; $f$ - feature; $t$ - threshold; |
| **2 if** *termination criterion is met* **then** |
| **3** $\quad$ Compute the prediction value at $\kappa$ using $S$; |
| **4 else** |
| **5** $\quad$ Choose an optimal split $(f, t)$; |
| **6** $\quad$ Create two child nodes $\kappa_{f>t}$ and $\kappa_{f\leq t}$ for $\kappa$; |
| **7** $\quad$ LearnDT($\kappa_{f>t}, S_{f>t}$); |
| **8** $\quad$ LearnDT($\kappa_{f\leq t}, S_{f\leq t}$); |
| **9 end** |

the test's outcome, we create two child nodes $\kappa_{f>t}$ and $\kappa_{f\leq t}$, split $S$ into two subsets $S_{f>t}$ and $S_{f\leq t}$, and finally continue the learning process with $(\kappa_{f>t}, S_{f>t})$ and $(\kappa_{f\leq t}, S_{f\leq t})$.

Given an impurity measure Impurity($S$), a split $(f, t)$'s impurity reduction is

$$\mathrm{IR}(f, t; S) := \mathrm{Impurity}(S) - \frac{|S_{f>t}|}{|S|}\mathrm{Impurity}(S_{f>t}) - \frac{|S_{f\leq t}|}{|S|}\mathrm{Impurity}(S_{f\leq t}). \quad (1)$$

Typically, the Gini impurity $G(S)$ and the entropy $H(S)$ are used as the impurity measure:

$$G(S) := 1 - q_+^2 - q_-^2 = 2q_+(1 - q_+), \qquad H(S) := -q_+ \ln q_+ - q_- \ln q_-, \quad (2)$$

where $q_+$ and $q_-$ are the proportions of positive and negative examples in $S$ respectively.

We will use $\mathrm{IR}_{\mathrm{Gini}}(f, t; S)$ and $\mathrm{IR}_{\mathrm{entropy}}(f, t; S)$ to denote the impurity reduction when using the Gini impurity and the entropy impurity respectively.

**PU learning**    We assume that the input $\boldsymbol{x} \in \mathbb{R}^d$ and label $y \in \{-1, +1\}$ follow an unknown joint distribution $p(\boldsymbol{x}, y) = p(\boldsymbol{x} \,|\, y)\, p(y)$. In the fully supervised case, the training data is generally

assumed to be independently sampled from $p(\boldsymbol{x}, y)$. In the PU setting, one common assumption on the data generation mechanism [11, 10, 16] is that the positive examples $P = \{\boldsymbol{x}_i^{(\mathsf{p})}\}_{i=1}^{n_{\mathsf{p}}}$ are sampled independently from the P marginal $p_{\mathsf{p}}(\boldsymbol{x}) := p(\boldsymbol{x} \,|\, y = 1)$, and the unlabeled examples $U = \{\boldsymbol{x}_i^{(\mathsf{u})}\}_{i=1}^{n_{\mathsf{u}}}$ are sampled independently from the marginal $p(\boldsymbol{x}) = p(\boldsymbol{x}, +1) + p(\boldsymbol{x}, -1)$. Clearly, we have $p(\boldsymbol{x}) = \pi \, p_{\mathsf{p}}(\boldsymbol{x}) + (1 - \pi) \, p_{\mathsf{n}}(\boldsymbol{x})$, where $\pi := p(y = +1)$ is the positive rate and $p_{\mathsf{n}}(\boldsymbol{x}) := p(\boldsymbol{x} \,|\, y = -1)$ is the N marginal.

The objective is to learn a score function $g : \mathbb{R}^d \to \mathbb{R}$ (which corresponds to a classifier that outputs $+1$ iff $g(\boldsymbol{x}) > 0$) such that it minimizes its expected risk

$$R(g) := \mathbb{E}_{(\boldsymbol{x}, y) \sim p(\boldsymbol{x}, y)} \, \ell(g(\boldsymbol{x}), y), \tag{3}$$

where $\ell : \mathbb{R} \times \{-1, +1\} \to \mathbb{R}$ is a loss function that gives the loss $\ell(v, y)$ incurred by predicting a score $v$ when the true label is $y$. We focus on the quadratic loss $\ell_{\mathrm{quad}}(v, y) = (1 - vy)^2$ , the logistic loss $\ell_{\mathrm{logistic}}(v, y) = \ln(1 + \exp(-vy))$ , the savage loss $\ell_{\mathrm{savage}}(v, y) = 4/(1 + \exp(vy))^2$ and the sigmoid loss $\ell_{\mathrm{sigmoid}}(v, y) = 1/(1 + \exp(vy))$ in this paper.

The risk can be written in terms of positive and unlabeled data as [11]

$$R(g) = \pi \, \mathbb{E}_{\boldsymbol{x} \sim p_{\mathsf{p}}(\boldsymbol{x})} \, \ell(g(\boldsymbol{x}), +1) + \mathbb{E}_{\boldsymbol{x} \sim p(\boldsymbol{x})} \, \ell(g(\boldsymbol{x}), -1) - \pi \, \mathbb{E}_{\boldsymbol{x} \sim p_{\mathsf{p}}(\boldsymbol{x})} \, \ell(g(\boldsymbol{x}), -1). \tag{4}$$

This gives the following unbiased PU-data based risk estimator

$$\widehat{R}_{\mathrm{uPU}}(g) := \sum_{\boldsymbol{x} \in P} w_{\mathsf{p}} \ell(g(\boldsymbol{x}), +1) - \sum_{\boldsymbol{x} \in P} w_{\mathsf{p}} \ell(g(\boldsymbol{x}), -1) + \sum_{\boldsymbol{x} \in U} w_{\mathsf{u}} \ell(g(\boldsymbol{x}), -1), \tag{5}$$

where $w_{\mathsf{p}} = \pi/n_{\mathsf{p}}$ and $w_{\mathsf{u}} = 1/n_{\mathsf{u}}$.

While the uPU risk estimator is unbiased, it has a negative component that provides extra incentive for the classifier to try hard to fit to the positive examples, thus potentially making the risk negative and leading to overfitting. The non-negative risk estimator

$$\widehat{R}_{\mathrm{nnPU}}(g) := \sum_{\boldsymbol{x} \in P} w_{\mathsf{p}} \ell(g(\boldsymbol{x}), +1) + \max \left\{ 0, \sum_{\boldsymbol{x} \in U} w_{\mathsf{u}} \ell(g(\boldsymbol{x}), -1) - \sum_{\boldsymbol{x} \in P} w_{\mathsf{p}} \ell(g(\boldsymbol{x}), -1) \right\} \tag{6}$$

alleviates this problem by forcing the sum of the negative term and the term defined on the unlabeled data to be non-negative, as the sum acts as an estimate for risk on negative examples

## 3 Recursive Greedy Risk Minimization

### 3.1 Decision trees for positive and negative data

We first introduce our recursive greedy risk minimization approach in the fully labeled case. Consider the empirical risk of $g : \mathbb{R}^d \to \mathbb{R}$ on a labeled training set $D = \{(\boldsymbol{x}_1, y_1), \ldots, (\boldsymbol{x}_n, y_n)\}$: $\widehat{R}(g) = \sum_{(\boldsymbol{x}, y) \in D} w \ell(g(\boldsymbol{x}), y)$, where $w = 1/|D|$. If $g$ predicts a constant $v$ on a subset $S = \{(\boldsymbol{x}, y)\}$ of the training examples, then the contribution to the total empirical risk is the *partial empirical risk* $\widehat{R}(v; S) := \sum_{(\boldsymbol{x}, y) \in S} w \ell(v, y)$. The optimal constant value prediction is $v_S^* = \mathrm{argmin}_{v \in \mathbb{R}} \widehat{R}^*(v; S)$ with a minimum partial empirical risk $\widehat{R}^*(S) := \widehat{R}^*(v_S^*; S)$.

If we switch from a constant value prediction rule to a decision stump that uses the split $(f, t)$, then the minimum partial empirical risk for such a decision stump is $\widehat{R}^*(S_{f>t}) + \widehat{R}^*(S_{f \leq t})$, thus the *risk reduction* for the split $(f, t)$ is

$$\mathrm{RR}(f, t; S) := \widehat{R}^*(S) - \widehat{R}^*(S_{f>t}) - \widehat{R}^*(S_{f \leq t}). \tag{7}$$

In the following, when we need to make the loss function explicit, we will use subscript to indicate that. For example, $\widehat{R}^*_{\mathrm{quad}}(S)$ and $\mathrm{RR}_{\mathrm{quad}}(f, t; S)$, are the minimum partial empirical risk for a constant-valued prediction on $S$ and the risk reduction for the split $(f, t)$ on $S$ when using the quadratic loss.

When we choose the $(f, t)$ with maximum risk reduction in Algorithm 1, we obtain a recursive greedy risk minimization algorithm that recursively zooms in to an input region with a constant prediction,

then improves the prediction rule in a greedy manner by replacing it with the decision stump with minimal empirical risk in that region.

The first main result shows that the Gini impurity reduction and entropy reduction of a split are just scaled versions of the risk reductions when using the quadratic loss and the logistic loss respectively. This implies that the standard impurity reduction based decision learning algorithm is in fact performing recursive greedy risk minimization. All proofs are in the appendix.

**Theorem 1.** *(a) For any $S \subseteq D$, we have $\widehat{R}^*_{\mathrm{quad}}(S) = 2|S|wG(S)$. As a consequence, for any $S \subseteq D$ and any split $(f, t)$ [4],*

$$\mathrm{RR}_{\mathrm{quad}}(f, t; S) = 2|S|w\mathrm{IR}_{\mathrm{gini}}(f, t; S). \tag{9}$$

*(b) For any $S \subseteq D$, we have $\widehat{R}^*_{\mathrm{logistic}}(S) = |S|wH(S)$. As a consequence, for any $S \subseteq D$ and any split $(f, t)$,*

$$\mathrm{RR}_{\mathrm{logistic}}(f, t; S) = |S|w\mathrm{IR}_{\mathrm{entropy}}(f, t; S). \tag{10}$$

### 3.2 Decision trees for positive and unlabeled data

We are now ready to introduce a recursive greedy risk minimization approach to PU learning. We do this by making two changes to Algorithm 1: the fully labeled dataset $S$ is replaced by a set $P' \subseteq P$ of positive examples and a set $U' \subseteq U$ of unlabeled examples, and the split $(f, t)$ is chosen to optimize a PU version of the risk reduction.

**uPU risk reduction** We first consider the simpler case of the uPU estimator. Similarly to the fully labeled case, if $g$ predicts a constant $v$ on $P'$ and $U'$, then the contribution to the total empirical uPU risk is

$$\widehat{R}_{\mathrm{uPU}}(v; P', U') := \sum_{\boldsymbol{x} \in P'} w_{\mathsf{p}}\ell(v, +1) - \sum_{\boldsymbol{x} \in P'} w_{\mathsf{p}}\ell(v, -1) + \sum_{\boldsymbol{x} \in U'} w_{\mathsf{u}}\ell(v, -1).$$

The optimal constant value prediction is $v^*_{P', U'} = \mathrm{argmin}_{v \in \mathbb{R}} \widehat{R}_{\mathrm{uPU}}(v; P', U')$ with a minimum empirical risk $\widehat{R}^*_{\mathrm{uPU}}(P', U') := \widehat{R}_{\mathrm{uPU}}(v^*_{P', U'}; P', U')$. The uPU *risk reduction* for the split $(f, t)$ on $(P', U')$ is

$$\mathrm{RR}_{\mathrm{uPU}}(f, t; P', U') := \widehat{R}^*_{\mathrm{uPU}}(P', U') - \widehat{R}^*_{\mathrm{uPU}}(P'_{f>t}, U'_{f>t}) - \widehat{R}^*_{\mathrm{uPU}}(P'_{f\le t}, U'_{f\le t}). \tag{8}$$

The following result shows that the optimal risk $\widehat{R}^*_{\mathrm{uPU}}(P', U')$ has a closed-form formula and is thus efficiently computable. As a consequence, the uPU risk reduction for a split $(f, t)$ is efficiently computable.

**Proposition 1.** *Consider arbitrary $P' \subseteq P$ and $U' \subseteq U'$. Let $W_{\mathsf{p}} = |P'|w_{\mathsf{p}}$, $W_{\mathsf{n}} = |U'|w_{\mathsf{u}} - |P'|w_{\mathsf{p}}$, and $v^* := \frac{W_{\mathsf{p}}}{W_{\mathsf{p}} + W_{\mathsf{n}}}$.*

*(a) When using the quadratic loss, we have*

$$\widehat{R}^*_{\mathrm{uPU}}(P', U') = \begin{cases} -\infty, & v^* = +\infty, \\ 4(W_{\mathsf{p}} + W_{\mathsf{n}})v^*(1 - v^*), & \text{otherwise.} \end{cases} \tag{11}$$

*(b) When using the logistic loss, we have*

$$\widehat{R}^*_{\mathrm{uPU}}(P', U') = \begin{cases} (W_{\mathsf{p}} + W_{\mathsf{n}})(-v^* \ln(v^*) - (1 - v^*)\ln(1 - v^*)), & 0 < v^* < 1, \\ 0, & v^* \in \{0, 1\}, \\ -\infty, & v^* > 1. \end{cases} \tag{12}$$

Intuitively, if $P' \subseteq P$ and $U' \subseteq U$ are the examples in a node $\kappa$ in the decision tree, then the weights $W_{\mathsf{p}}$ and $W_{\mathsf{n}}$ serve as unbiased estimates of the probabilities $p(\boldsymbol{x} \in \kappa, y = 1)$ and $p(\boldsymbol{x} \in \kappa, y = -1)$ respectively. See Appendix C for details. Thus $v^* = \frac{W_{\mathsf{p}}}{W_{\mathsf{p}} + W_{\mathsf{n}}}$ serves as an estimate of the probability of positive examples. Note that $W_{\mathsf{p}} \in [0, 1]$, while $W_{\mathsf{n}} \le 1$ is unusual in that it contains a negative component and hence can be negative. Consequently, $v^*$ is non-negative, but it can be larger than 1 though. In particular, if $W_{\mathsf{p}} + W_{\mathsf{n}} = |U'|w_{\mathsf{u}} = 0$, then $v^* = +\infty$.

nnPU **risk reduction**    For the case of the nnPU estimator, define

$$\widehat{R}_{\text{nnPU}}(v; P', U') := \sum_{\boldsymbol{x} \in P'} w_{\mathsf{p}} \ell(v, +1) + \max \left\{ 0, \sum_{\boldsymbol{x} \in U'} w_{\mathsf{u}} \ell(v, -1) - \sum_{\boldsymbol{x} \in P'} w_{\mathsf{p}} \ell(v, -1) \right\}.$$

We can then similarly define the risk reduction for the nnPU estimator.

The following result shows that in the nnPU case, the optimal risk $\widehat{R}^*_{\text{uPU}}(P', U')$ also has a closed-form formula and is thus efficiently computable.

**Proposition 2.** *Using the same notations as in Proposition 1, we have the following results.*

*(a) When using the quadratic loss, we have*

$$\widehat{R}^*_{\text{nnPU}}(P', U') = \begin{cases} 0, & v^* > 1 \\ 4(W_{\mathsf{p}} + W_{\mathsf{n}})v^* (1 - v^*), & \text{otherwise.} \end{cases} \tag{14}$$

*(b) When using the logistic loss, we have*

$$\widehat{R}^*_{\text{nnPU}}(P', U') = \begin{cases} (W_{\mathsf{p}} + W_{\mathsf{n}}) \left( -v^* \ln(v^*) - (1 - v^*) \ln(1 - v^*) \right), & 0 < v^* < 1 \\ 0, & \text{otherwise.} \end{cases} \tag{15}$$

It is important to note that while in the uPU case, if $P'$ is partitioned into $P'_1$ and $P'_2$, and $U'$ into $U'_1$ and $U'_2$, then we have $\widehat{R}_{\text{uPU}}(u; P', U') = \widehat{R}_{\text{uPU}}(u, P'_1, U'_1) + \widehat{R}_{\text{uPU}}(u, P'_2, U'_2)$, we only have $\widehat{R}_{\text{nnPU}}(u; P', U') \leq \widehat{R}_{\text{nnPU}}(u, P'_1, U'_1) + \widehat{R}_{\text{nnPU}}(u, P'_2, U'_2)$. Thus recursive greedy risk minimization minimizes an upper bound of the partial empirical risk rather than the partial empirical risk. This can be considered as a regularization mechanism. In fact, we observe nnPU risk to work better than the uPU risk in our experiments, which is consistent with the findings in [16].

**Optimizing the split**    With the above closed-form formula, we can then efficiently compute the risk reduction for a given split $(f, t)$ on $(P', U')$. While there are infinitely many possible $t$ values to choose from, note that they often split the dataset in the same way, and if $f$'s values split $\mathbb{R}$ into multiple intervals, we only need to consider $t_1 < t_2 < \ldots < t_m$, each chosen from an interval so that we cover all possible cases. While computing the risk reduction for each $(f, t_i)$ takes $O(|P'| + |U'|)$ time, we can efficiently get the risk reductions for all of $(f, t_1), \ldots, (f, t_m)$ in $O((|P'| + |U'|) \ln(|P'| + |U'|) + m)$ time, by sorting the examples according to their $f$ values, and going through the thresholds in a sorted order. See Appendix E for details.

**Invariance properties**    An interesting property of the greedy recursive learning approach is that it is robust to the choice of the loss function in the sense that different loss functions can lead to identical decision trees, as shown in the result below.

**Proposition 3.** *The $\widehat{R}^*(S)$ value is the same for the quadratic loss and the savage loss.*

Another interesting property is that the optimal prediction among $-1$ and $+1$ are often the same when we use different estimators and different loss functions.

**Proposition 4.** *If a node contains the examples $P' \subseteq P$ and $U' \subseteq U$, $v^*$ is as defined in Proposition 1, and the prediction is chosen from $\{-1, +1\}$ to minimize the empirical risk estimate on $P'$ and $U'$, then the optimal prediction is $2\mathbb{1}(v^* > 0.5) - 1$ for both the uPU and nnPU risk estimate and all the loss functions in Section 2. Intuitively, we predict $+1$ iff the estimated proportion of positive data at the current node is larger than 0.5.*

## 4   PU Extra Trees

With our PU decision tree algorithms, we can obtain a PU random forest algorithm by adapting the random forest algorithm [3] for positive and negative data: repeatedly obtain bootstrap samples for both the positive data and the unlabeled data, then train a PU decision tree for each sample with optimal split chosen over a randomly sampled subset of features each time. While choosing the optimal split from a subset of features makes training each decision tree faster, it is still the most expensive part in the tree construction process and can potentially be very slow on a large dataset. We describe a more efficient version that we used in our experiments below. In addition, we introduce a new feature importance score that directly measures a feature's contribution to risk minimization.

**PU ET (Extra Trees)**   We develop a more efficient random forest algorithm by using the randomization trick in the Extra Trees algorithm [13] to further reduce the computational cost for finding an optimal split below. Besides sampling only a subset of features, only one random threshold for each sampled feature is considered. We implemented a more general version which allows sampling multiple random thresholds for a sampled feature. A complete pseudocode for PU ET is given in Appendix H.

**Termination criteria**   Decision trees can overfit the training data if allowed to grow until each leaf has just one example. Learning is generally terminated early to alleviate overfitting. In our implementation, we terminate when all feature values are constant, when a maximum tree depth $d_{\max}$ is reached, when the node size falls below a threshold $s_{\min}$, or when the node is *pure*. A node is said to be pure if the impurity measure takes value $-\infty$ in the uPU setting, or value 0 in the nnPU setting.

**Risk reduction importance**   Our feature importance, called *risk reduction importance*, is the total risk reduction for a feature across all nodes in all the trees. The risk reduction importance of a feature $f$ is defined as $\sum_{\kappa \in K_f} \text{RR}(f_\kappa, t_\kappa; P_\kappa, U_\kappa)$, where $K_f$ is the set of nodes using $f$ as a splitting feature, $(f_\kappa, t_\kappa)$ is the split for node $\kappa$, and $(P_\kappa, U_\kappa)$ is the set of PU data at $\kappa$. This can be averaged across multiple trees in a random forest. The risk reduction importance in PU learning is similar to the Gini importance as defined in, for example, [4], whereby the size of the nodes is taken into account. This is beneficial as otherwise features often used in splitting small datasets may appear very important as compared to features used in splitting large datasets, even though it does not contribute as much to reducing the risk and improving classification performance. To demonstrate the effect of taking data size into account, a normalized risk reduction can be defined by dividing the risk reduction by the total weight of the examples.

## 5   Experiments

In this section we compare PU Extra Trees with several other PU learning methods. Selected tree-based methods include NaivePUET whereby ET is trained on the PU dataset by simply treating U data as N data; PUBaggingET which uses PUBagging from [23] with ET base classifier; and SupervisedET where ET is trained on the original fully labeled dataset (expected to be an upper bound for tree based PU methods). To further study the effectiveness of our PU Extra Trees algorithm we also compare against neural network based methods including a baseline method uPU [11, 10] and two state of the art methods nnPU [16] and Self-PU [6]. We ran experiments on heterogeneous devices. In particular, random forests were trained using 32GB RAM and one of Intel i7-10700, Intel i7-11700 or AMD Epyc 7702p CPU. Neural networks were trained on one of NVIDIA RTX A4000 or NVIDIA RTX A6000 GPU due to the lack of identical devices. While the running times are not always comparable and thus not provided in the main text, experiments show that our current implementation of PU ET takes seconds to train a single-tree random forest on modest hardware. See Appendix I for details.

**Datasets**   We consider a selection of common datasets for classification from LIBSVM [5], as well as MNIST digits [19], the intrusion detection dataset UNSW-NB15 [25] and CIFAR-10 [17] to demonstrate the versatility of our method. Table 1 is a summary of the benchmark datasets.

Table 1: Benchmark datasets. *: random 80%-20% train-test split was used as no train-test splits were provided.

| Name | # Train | # Test | # Feature | $\pi$ |
|---|---|---|---|---|
| mushrooms* | 6499 | 1625 | 112 | 0.52 |
| 20News | 11 314 | 7 532 | 300 | 0.56 |
| covtype.binary* | 464 809 | 116 203 | 54 | 0.51 |
| epsilon | 400 000 | 100 000 | 2 000 | 0.5 |
| MNIST | 60 000 | 10 000 | 784 | 0.5 |
| CIFAR-10 | 50 000 | 10 000 | 3 072 | 0.4 |
| UNSW-NB15 | 175 340 | 82 331 | 39 | 0.68 |

The 20News, epsilon, MNIST and CIFAR-10 datasets were processed in the same way as in [16] for consistency. Definitions of labels ('positive' vs 'negative') are as follows: For 20News, 'alt.,

comp., misc., rec.' vs 'sci., soc., talk.'; for MNIST, '0,2,4,6,8' vs '1,3,5,7,9'; for CIFAR-10, 'airplane, automobile, ship, truck' vs 'bird, cat, deer, dog, frog, horse'; for UNSW-NB15, all attack types make up the P class and the benign data makes up the N class; for mushroom and covtype, the most prominent class makes up the P class; and epsilon dataset is provided with P and N classes. GloVe pre-trained word embeddings [27] were used for 20News with average pooling over each document. Each feature in the covtype and UNSW-NB15 datasets was scaled between 0 and 1 when using uPU and nnPU as there is a large mismatch in scales between the features which seemed to significantly reduce performance of the methods. No such scaling of the data is required for PU ET as performance does not seem to be affected in such cases.

To convert the PN data to PU data we follow the experimental setup in [16] whereby 1000 positive examples are randomly sampled to form the P set, and the entire dataset is used for the U set. In practice it may be the case that $\pi$ is not known a-priori, in which case there are methods for estimating $\pi$ using only PU data, for example, in [7], or [28].

**Model hyperparameters** Following common practice [26, 13], the default hyperparameters for PU ET are: 100 trees, no explicit restriction on the maximum tree depth, sample $F = \lceil \sqrt{d} \rceil$ features out of a total of $d$ features and sample $T = 1$ threshold value when computing an optimal split. The size of the bootstrap sample for PUBaggingET is set to the default value of the dataset size.

The architectures for the neural networks used in uPU, nnPU and Self-PU were copied from [16] for the 20News, epsilon, MNIST and CIFAR-10 datasets. A 6 layer MLP with ReLU was used for MNIST, Covtype, Mushroom and UNSW-NB15; a similar model was used for epsilon while the activation was replaced with Softsign; a 5 layer MLP with Softsign was used for 20News. All hidden layers had width 300 for the MLPs. The model for CIFAR-10 was the 13 layer CNN: (32*32*3,1)-[C(3*3,96,1)]*2-C(3*3,96,2)-[C(3*3,192,1)]*2-C(3*3,192,2)-C(3*3,192,1)-C(1*1,192,1)-C(1*1,10,1)-1000-1000-1, where the input is a 32*32 RGB image, C(3*3,96,1) means 96 channels of 3*3 convolutions with stride 1 followed by ReLU, [.]*2 means there are two such layers. For the remaining datasets we used a 6-layer MLP with ReLU [18, 14] (more specifically, $d$-300-300-300-300-1). For each dataset the neural networks were trained for 200 epochs. The batch size, learning rate, use of batch-norm (for datasets not included in [16]), weight decay and choice of optimiser were tuned for each dataset.

It is noteworthy that we spend little effort to tune PU ET hyperparameters and simply use the same hyperparameters on all datasets. In addition, PU ET seems to be robust to hyperparameter choice in that the default selection tends to result in strong predictive performance with decreased training times. See Appendix J for results from hyperparameter tuning experiment. On the other hand, neural nets are sensitive to hyperparameter choice, and significant tuning is often needed for each dataset.

## 5.1 Classification performance

To take into account of the randomness in the learning algorithms, we train each model five times on the same dataset, then report the mean and standard deviation of the accuracy and the F-score. Results for uPU, nnPU and Self-PU were reproduced to the best of our ability, particularly on the datasets that were included in the original results section of the respective papers.

**Effect of loss function and risk estimator on PU ET** We first evaluate the effect of different loss functions and risk estimators for PU ET. The results are shown in Table 2 and Table 3. PU ET with nonnegative risk estimator seems to perform about as well using either quadratic or logistic loss. Unbiased risk estimator seems to lead to overfitting, but less so with logistic loss. Logistic loss leads to better performance than quadratic loss when using unbiased risk estimator. Note that (11) implies that quadratic loss encourages aggressive splitting until all examples are positive as a loss of $-\infty$ can be achieved in this case, while (12) implies that logistic loss has a less aggressive splitting behavior as a loss of $-\infty$ is achieved when the total weight of positive examples exceeds the total weight of unlabeled examples. As discussed previously, once we achieve an impurity value of $-\infty$ in uPU, we stop splitting. The more aggressive splitting behavior of quadratic loss leads to deeper trees that tend to overfit more than the logistic loss in general. This is consistent with results from additional experiments in Appendix K.

**Comparison of PU ET with other tree-based PU learning methods** We now compare our PU ET algorithm with several tree-based baselines to further study the effectiveness of our novel PU tree

Table 2: Accuracy mean% (sd) on the test set for PU ET using various PU data based risk estimators.

| Dataset | Unbiased risk estimator | | Nonnegative risk estimator | |
|---|---|---|---|---|
| | Quadratic Loss | Logistic Loss | Quadratic Loss | Logistic Loss |
| Epsilon | 50.04 (0.00) | 50.67 (0.08) | 57.39 (0.76) | 57.83 (0.70) |
| 20News | 43.64 (0.05) | 73.44 (0.87) | 83.34 (0.22) | 83.34 (0.31) |
| Covtype | 48.72 (0.00) | 72.63 (0.87) | 76.51 (0.52) | 75.63 (0.52) |
| Mushroom | 0.607 (0.48) | 99.02 (0.32) | 99.70 (0.24) | 99.36 (0.48) |
| MNIST | 50.74 (0.0) | 89.05 (0.61) | 93.60 (0.39) | 94.01 (0.19) |
| CIFAR-10 | 60.0.(0.01) | 75.30 (0.66) | 79.74 (0.37) | 79.86 (0.39) |
| UNSW-NB15 | 47.18 (0.15) | 83.89 (1.03) | 82.24 (0.86) | 81.51 (1.58) |

Table 3: F mean% (sd) on the test set for PU ET using various PU data based risk estimators.

| Dataset | Unbiased risk estimator | | Nonnegative risk estimator | |
|---|---|---|---|---|
| | Quadratic Loss | Logistic Loss | Quadratic Loss | Logistic Loss |
| Epsilon | 0.00 (0.00) | 4.18 (0.59) | 39.52 (1.19) | 40.25 (1.38) |
| 20News | 0.43 (0.17) | 72.65 (1.11) | 85.33 (0.22) | 85.26 (0.28) |
| Covtype | 0.11 (0.01) | 68.44 (1.25) | 75.19 (0.37) | 74.20 (0.58) |
| Mushroom | 39.02 (1.19) | 99.04 (0.32) | 99.71 (0.24) | 99.38 (0.47) |
| MNIST | 0.00 (0.00) | 88.06 (0.74) | 93.49 (0.42) | 93.90 (0.19) |
| CIFAR-10 | 0.02 (0.04) | 61.33 (1.44) | 71.31 (0.43) | 71.67 (0.53) |
| UNSW-NB15 | 7.97 (0.47) | 86.33 (0.87) | 85.65 (0.59) | 85.17 (1.15) |

learning method. The results are shown in Table 4 and Table 5. The results support the effectiveness of our algorithm. Our PU ET with nnPU risk estimator and quadratic loss significantly outperforms both NaivePUET and PUBaggingET, particularly in terms of F-scores. PU ET shows strong performance even as compared to SupervisedET, though often using a small proportion of the positive labels only.

Table 4: Accuracy mean% (sd) on the test set for various tree based methods. [†]: Original dataset with full supervision was used during training.

| Dataset | SupervisedET[†] | PU ET | NaivePUET | PUBaggingET |
|---|---|---|---|---|
| Epsilon | 73.55 (0.08) | 57.39 (0.76) | 50.04 (0.00) | 50.04 (0.00) |
| 20news | 85.39 (0.12) | 83.34 (0.22) | 43.63 (0.05) | 43.75 (0.06) |
| Covtype | 95.90 (0.02) | 76.51 (0.52) | 48.71 (0.00) | 48.71 (0.0) |
| Mushroom | 100 (0.00) | 99.70 (0.24) | 53.85 (0.32) | 62.76 (0.84) |
| MNIST | 98.11 (0.05) | 93.60 (0.39) | 50.74 (0.00) | 50.74 (0.00) |
| CIFAR-10 | 85.02 (0.19) | 79.74 (0.37) | 60.00 (0.00) | 60.0 (0.00) |
| UNSW-NB15 | 86.57 (0.05) | 82.24 (0.86) | 44.95 (0.01) | 44.97 (0.02) |

Table 5: F mean% (sd) on the test set for PU ET using various PU data based risk estimators. [†]: Original dataset with full supervision was used during training.

| Dataset | SupervisedET[†] | PU ET | NaivePUET | PUBaggingET |
|---|---|---|---|---|
| Epsilon | 72.96 (0.08) | 39.52 (1.19) | 0.00 (0.00) | 0.00 (0.00) |
| 20news | 87.40 (0.11) | 85.33 (0.22) | 0.38 (0.19) | 0.82 (0.18) |
| Covtype | 95.97 (0.02) | 75.19 (0.37) | 0.05 (0.01) | 0.06 (0.01) |
| Mushroom | 100 (0.00) | 99.71 (0.24) | 19.68 (2.04) | 45.3 (0.84) |
| MNIST | 98.09 (0.05) | 93.49 (0.42) | 0.00 (0.00) | 0.00 (0.00) |
| CIFAR-10 | 80.30 (0.24) | 71.31 (0.43) | 0.01 (0.02) | 0.02 (0.02) |
| UNSW-NB15 | 88.96 (0.04) | 85.65 (0.59) | 0.08 (0.06) | 0.11 (0.06) |

**Comparison of PU ET with neural network methods**  We compare PU ET with nnPU risk estimator and quadratic loss against uPU neural nets, nnPU neural nets and Self-PU. The results are shown in Table 6 and Table 7. PU ET shows strong performance in terms of both accuracy and F score on a wide variety of datasets compared to neural network based models.

Table 6: Accuracy% mean (sd) on the test set for PU ET with nonnegative risk estimator and quadratic loss and three neural network based PU learning methods. *: Self-PU was run without self-calibration due to limitations with available hardware.

| Dataset | PU ET | Neural Network | | |
| | | uPU | nnPU | Self-PU |
| --- | --- | --- | --- | --- |
| Epsilon | 57.39 (0.76) | 61.79 (2.57) | 73.80 (0.49) | 50.05 (0.0) |
| 20News | 83.34 (0.22) | 44.95 (1.09) | 74.11 (3.65) | 62.87 (2.89) |
| Covtype | 76.51 (0.52) | 72.38 (1.23) | 71.64 (0.66) | 54.67 (0.40) |
| Mushroom | 99.70 (0.24) | 98.09 (1.03) | 99.04 (0.54) | 99.25 (0.61) |
| MNIST | 93.60 (0.39) | 55.95 (1.17) | 93.08 (0.46) | 93.13 (0.41) |
| CIFAR-10 | 79.74 (0.37) | 62.68 (2.95) | 81.87 (04.20) | 88.22 (0.51)* |
| UNSW-NB15 | 82.24 (0.86) | 76.62 (0.01) | 76.63 (0.02) | 76.73 (0.18) |

Table 7: F mean% (sd) on the test set for PU ET with nonnegative risk estimator and quadratic loss and three neural network based PU learning methods. *: Self-PU was run without self-calibration due to limitations with available hardware.

| Dataset | PU ET | Neural Network | | |
| | | uPU | nnPU | Self-PU |
| --- | --- | --- | --- | --- |
| Epsilon | 39.52 (1.19) | 43.44 (8.14) | 71.53 (0.46) | 0.00 (0.00) |
| 20News | 85.33 (0.22) | 12.95 (3.82) | 78.05 (3.81) | 71.78 (6.85) |
| Covtype | 75.19 (0.37) | 69.56 (1.77) | 68.29 (0.93) | 0.501 (2.55) |
| Mushroom | 99.71 (0.24) | 98.13 (1.02) | 99.06 (0.53) | 99.27 (0.60) |
| MNIST | 93.49 (0.42) | 19.13 (3.82) | 92.77 (0.60) | 92.96 (0.47) |
| CIFAR-10 | 71.31 (0.43) | 32.43 (25.03) | 76.01 (7.62) | 85.51 (0.60)* |
| UNSW-NB15 | 85.65 (0.59) | 82.47 (0.01) | 82.48 (0.02) | 80.18 (0.28) |

## 5.2 Feature Importance

Interpretability is a desirable property of machine learning algorithm, but neural networks are often hard to interpret. In contrast, tree based methods are generally easier to interpret. We illustrate our risk reduction importances of both PU ET and PN (positive-negative) ET on the UNSW-NB15 and the MNIST datasets (lighter color indicates a higher score) in Figure 1 and Figure 2 respectively .

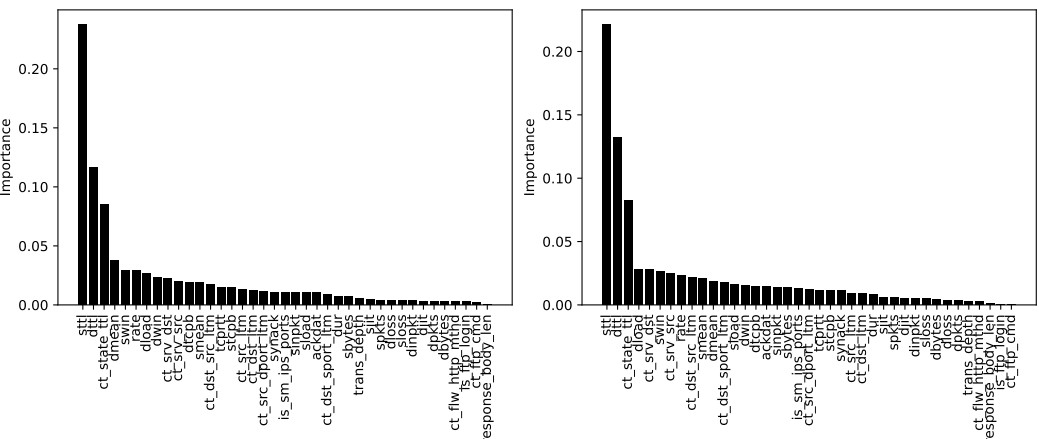

Figure 1: Risk reduction importances on UNSW-NB15. Left: PU ET. Right: PN ET.

For MNIST, the plot of the importance scores for PU ET for each digit often suggests the shape of the digit. On the other hand, the Appendix provides some additional figures that show that the normalized risk reduction importance makes many more pixels more important. This observation is consistent with our discussion in Section 4. Interestingly, the importances for PU ET appear to be very similar to those for PN ET (visually similar importance maps for MNIST, and qualitatively

similar importance values which give similar feature rankings for UNSW-NB15). This suggests that the PU model is likely quite similar to the PN model.

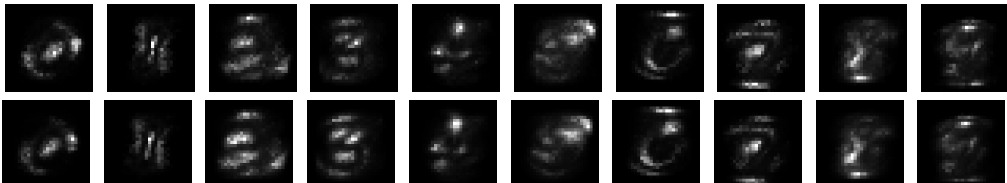

Figure 2: Risk reduction importances learnt from ET classifier on MNIST digits. Top: PU ET with nonnegative risk estimator and quadratic loss. Bottom: PN ET with Gini impurity.

# 6    Related Work

PU learning was initially studied in [9, 8, 21], and many approaches have been developed based on different assumptions on the data generation mechanism.

The two-step approach discussed in the introduction have been adopted in many papers with numerous instantiations. We refer the readers to [2] for a comprehensive list.

Another important line of work, which is most related to our work and has achieved the state-of-the-art performance recently, minimizes a risk estimate based on a weighted dataset of positive and negative examples obtained by transforming the PU dataset [20, 12, 11, 16]. Our work follows the risk minimization approach, but we develop general risk minimization algorithms for learning tree models, that allows exploiting the under-explored potential of random forests.

Numerous works have been done on tree-based methods, and random forest algorithms are among the most successful ones [3, 13, 1]. In the PU learning literature, PU decision trees were explored in early works [8, 21], and PU bagging [23] has been explored for SVMs [23] and decision trees [29]. All of these approaches do not aim to directly minimize the risk.

Our recursive greedy risk minimization framework gives a new interpretation to existing tree learning algorithms for positive and negative data, and allows us to develop new algorithms for positive and unlabeled data. In addition, our approach allows us to formulate a new feature importance score that directly measures a feature's contribution to risk minimization.

# 7    Conclusions

Random forests perform very well on fully supervised learning methods, but its potential in PU learning has been under-explored due to the strong emphasis on neural networks and the lack of a principled method for learning decision trees from PU data. We fill this gap in this paper. We first consider learning from positive and negative data, and develop a recursive greedy risk minimization approach for learning decision trees, and show that with suitable choice of loss functions, its instantiations are equivalent to standard impurity-based decision tree learning algorithms. We extend this approach to the PU setting, and develop a very efficient random forest algorithm which requires little hyperparameter and yet strong performance against the state-of-the-art PU learning algorithms. Our approach supports a new feature importance score that directly measures a feature's contribution to risk mnimization.

Our work opens up some new opportunities to build tree models. First, decision trees can easily overfit the training set, and various heuristics have been used in practice to alleviate overfitting. Our recursive greedy risk minimization framework provides an alternative way to control the size of the learned decision by always expanding the node with the largest risk reduction until the risk reduction is smaller than a threshold or when the tree size is larger than a threshold. Second, we focused on quadratic loss and logistic loss in our experiments, but we can also study alternative losses such as the hinge loss, or double hinge loss.

## Acknowledgments and Disclosure of Funding

The authors would like to thank Dr Miao Xie from AARNet for his valuable suggestions. This work was funded by The University of Queensland Cyber Security Seed Funding.

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
