# A  Proof of Theorem 1

**Theorem 1.** *(a) For any $S \subseteq D$, we have $\widehat{R}^*_{\mathrm{quad}}(S) = 2|S|wG(S)$. As a consequence, for any $S \subseteq D$ and any split $(f, t)$ [4],*

$$\mathrm{RR}_{\mathrm{quad}}(f, t; S) = 2|S|w\mathrm{IR}_{\mathrm{gini}}(f, t; S). \tag{9}$$

*(b) For any $S \subseteq D$, we have $\widehat{R}^*_{\mathrm{logistic}}(S) = |S|wH(S)$. As a consequence, for any $S \subseteq D$ and any split $(f, t)$,*

$$\mathrm{RR}_{\mathrm{logistic}}(f, t; S) = |S|w\mathrm{IR}_{\mathrm{entropy}}(f, t; S). \tag{10}$$

*Proof.* (a) This result is proven in [4].

(b) By grouping the positive and negative examples, we have

$$\widehat{R}_{\mathrm{logistic}}(v; S) = \sum_{(\boldsymbol{x}, y) \in S} w \ln(1 + e^{-vy}) = |S|wq_+ \ln(1 + e^{-v}) + |S|wq_- \ln(1 + e^{v}).$$

The minimizer satisfies

$$\frac{d\widehat{R}_{\mathrm{logistic}}}{dv} = |S|wq_+ \frac{-e^{-v}}{1 + e^{-v}} + |S|wq_- \frac{e^{v}}{1 + e^{v}} = 0.$$

Solving the equation, we have

$$v^*_S = \ln(q_+ / q_-),$$

or $e^{v^*_S} = q_+ / q_-$. Hence we have

$$\widehat{R}^*_{\mathrm{logistic}}(S) = |S|wq_+ \ln\left(1 + \frac{q_-}{q_+}\right) + |S|wq_- \ln\left(1 + \frac{q_+}{q_-}\right) = |S|wH(S).$$

$\square$

# B  Proof of Proposition 1

**Proposition 1.** *Consider arbitrary $P' \subseteq P$ and $U' \subseteq U'$. Let $W_{\mathsf{p}} = |P'|w_{\mathsf{p}}$, $W_{\mathsf{n}} = |U'|w_{\mathsf{u}} - |P'|w_{\mathsf{p}}$, and $v^* := \frac{W_{\mathsf{p}}}{W_{\mathsf{p}} + W_{\mathsf{n}}}$.*

*(a) When using the quadratic loss, we have*

$$\widehat{R}^*_{\mathrm{uPU}}(P', U') = \begin{cases} -\infty, & v^* = +\infty, \\ 4(W_{\mathsf{p}} + W_{\mathsf{n}})v^*(1 - v^*), & \text{otherwise.} \end{cases} \tag{11}$$

*(b) When using the logistic loss, we have*

$$\widehat{R}^*_{\mathrm{uPU}}(P', U') = \begin{cases} (W_{\mathsf{p}} + W_{\mathsf{n}})\left(-v^* \ln(v^*) - (1 - v^*)\ln(1 - v^*)\right), & 0 < v^* < 1, \\ 0, & v^* \in \{0, 1\}, \\ -\infty, & v^* > 1. \end{cases} \tag{12}$$

Both the set $P' \subseteq P$ and the set $U' \subseteq U$ can possibly be empty, thus $W_{\mathsf{p}} = |P'|w_{\mathsf{p}} \geq 0$, $W_{\mathsf{p}} + W_{\mathsf{n}} = |U'|w_{\mathsf{u}} \geq 0$, and $W_{\mathsf{n}} = |U'|w_{\mathsf{u}} - |P'|w_{\mathsf{p}}$ can be negative, 0, or positive. Note that we do not consider the case that $P'$ and $U'$ are both empty, as this only happens when no examples are allocated to a node, which does not arise when we construct decision trees.

*Proof.* (a) For the quadratic loss, we have

$$\widehat{R}_{\mathrm{uPU}}(v; P', U') = \sum_{x \in P'} w_{\mathsf{p}}(1 - v)^2 - \sum_{x \in P'} w_{\mathsf{p}}(1 + v)^2 + \sum_{x \in U'} w_{\mathsf{u}}(1 + v)^2$$
$$= W_{\mathsf{p}}(1 - v)^2 + W_{\mathsf{n}}(1 + v)^2.$$

If $W_p + W_n \neq 0$, then the partial empirical risk is a convex quadratic function with a unique minimizer satisfying

$$\frac{d\widehat{R}_{\mathrm{uPU}}}{dv} = -2W_p(1 - v^*_{P',U'}) + 2W_n(1 + v^*_{P',U'}) = 0.$$

Solving the equation, we have

$$v^*_{P',U'} = 2v^* - 1.$$

Therefore, we have

$$\begin{aligned}
\widehat{R}^*_{\mathrm{uPU}}(P', U') &= (W_p + W_n)v^*(1 - 2v^* + 1)^2 + (W_p + W_n)(1 - v^*)(1 + 2v^* - 1)^2 \\
&= 4(W_p + W_n)v^*(1 - v^*).
\end{aligned}$$

If $W_p + W_n = 0$, then $\widehat{R}_{\mathrm{uPU}}(v; P', U') = -2W_p v + W_p + 2W_n v + W_n$. This gives $v^*_{P',U'} = +\infty$, and $\widehat{R}^*_{\mathrm{uPU}}(P', U') = -\infty$.

(b) For the logistic loss, we have

$$\begin{aligned}
\widehat{R}_{\mathrm{uPU}}(v; P', U') &= \sum_{\boldsymbol{x} \in P'} w_p \ln(1 + e^{-v}) - \sum_{\boldsymbol{x} \in P'} w_p \ln(1 + e^{v}) + \sum_{\boldsymbol{x} \in U'} w_u \ln(1 + e^{v}) \\
&= W_p \ln(1 + e^{-v}) + W_n \ln(1 + e^{v}).
\end{aligned}$$

The derivative of the partial empirical risk wrt $v$ is

$$\frac{d\widehat{R}_{\mathrm{uPU}}}{dv} = \frac{W_n e^v - W_p}{1 + e^v}.$$

By inspecting how the sign of the derivative changes as $v$ changes, we can obtain

$$v^*_{P',U'} = \begin{cases} \ln(W_p/W_n), & W_n > 0, W_p > 0, \\ -\infty, & W_n > 0, W_p = 0, \\ +\infty, & W_n \leq 0. \end{cases} \tag{13}$$

with

$$\widehat{R}^*_{\mathrm{uPU}}(P', U') = \begin{cases} (W_p + W_n)(-v^* \ln(v^*) - (1 - v^*) \ln(1 - v^*)), & 0 < v^* < 1, \\ 0, & v^* \in \{0, 1\}, \\ -\infty, & v^* > 1. \end{cases}$$

$\square$

## C  Unbiased Estimates of Proportions

The weight $W_p$ is an unbiased estimate of the probability $p(\boldsymbol{x} \in \kappa, y = 1) = \pi p(\boldsymbol{x} \in \kappa \,|\, y = 1)$ since $\pi |P'|/|P| = |P'| w_p = W_p$ estimates the probability $\pi p(\boldsymbol{x} \in \kappa \,|\, y = 1)$ without bias.

To show that $W_n$ is an unbiased estimate of the probability $p(\boldsymbol{x} \in \kappa, y = -1)$, first note that for any $\boldsymbol{x}$ we have $(1 - \pi)p_n(\boldsymbol{x}) = p(\boldsymbol{x}) - \pi p_p(\boldsymbol{x})$. Integrating over $\boldsymbol{x} \in \kappa$, we have $(1 - \pi)p(\boldsymbol{x} \in \kappa \,|\, y = -1) = p(\boldsymbol{x} \in \kappa) - \pi p(\boldsymbol{x} \in \kappa \,|\, y = 1)$, or equivalently, $p(\boldsymbol{x} \in \kappa, y = -1) = p(\boldsymbol{x} \in \kappa) - \pi p(\boldsymbol{x} \in \kappa \,|\, y = 1)$. Now, $p(\boldsymbol{x} \in \kappa)$ is unbiasedly estimated by $|U'| w_u$, and $\pi p(\boldsymbol{x} \in \kappa \,|\, y = 1)$ is unbiasedly estimated by $|P'| w_p$.

## D  Proof of Proposition 2

**Proposition 2.** *Using the same notations as in Proposition 1, we have the following results.*

*(a) When using the quadratic loss, we have*

$$\widehat{R}^*_{\mathrm{nnPU}}(P', U') = \begin{cases} 0, & v^* > 1 \\ 4(W_p + W_n)v^*(1 - v^*), & \text{otherwise.} \end{cases} \tag{14}$$

*(b) When using the logistic loss, we have*

$$\widehat{R}^*_{\mathrm{nnPU}}(P', U') = \begin{cases} (W_p + W_n)(-v^* \ln(v^*) - (1 - v^*) \ln(1 - v^*)), & 0 < v^* < 1 \\ 0, & \text{otherwise.} \end{cases} \tag{15}$$

*Proof.* To find the minimal partial nnPU empirical risk, we begin by computing the derivative of $\widehat{R}_{\mathrm{nnPU}}$ with respect to $v$:

$$\widehat{R}_{\mathrm{nnPU}}(v; P', U') = \sum_{\boldsymbol{x} \in P'} w_{\mathsf{p}}(1-v)^2 + \max\left\{0, \sum_{\boldsymbol{x} \in U'} w_{\mathsf{u}}(1+v)^2 - \sum_{\boldsymbol{x} \in P'} w_{\mathsf{p}}(1+v)^2\right\}$$

$$= W_{\mathsf{p}}(1-v)^2 + \max\{0, W_{\mathsf{n}}(1+v)^2\},$$

$$\frac{\mathrm{d}\widehat{R}_{\mathrm{nnPU}}}{\mathrm{d}v} = 2W_{\mathsf{p}}(v-1) + 2\max\{0, W_{\mathsf{n}}\}(v+1).$$

First consider the case when $v^* > 1$, then either $\frac{W_{\mathsf{p}}}{W_{\mathsf{p}}+W_{\mathsf{n}}} > 1$ or $v^* = +\infty$. The former suggests that $W_{\mathsf{n}} < 0$, and hence the minimizer will be 1. In the latter we have $W_{\mathsf{n}} = -W_{\mathsf{p}} \leq 0$, again giving us a minimizer of 1. On the other hand if $0 \leq v^* < 1$, then we have $W_{\mathsf{p}} \geq 0$ and $W_{\mathsf{n}} > 0$ and the minimizer will be the solution to $0 = \mathrm{d}\widehat{R}_{\mathrm{nnPU}}/\mathrm{d}v$.

The minimizer can thus be written succinctly as

$$\operatorname*{argmin}_{v \in \mathbb{R}} \widehat{R}_{\mathrm{nnPU}}(v; P', U') = \begin{cases} 1, & v^* > 1, \\ 2v^* - 1, & \text{otherwise,} \end{cases}$$

with minimum partial empirical risk

$$\widehat{R}^*_{\mathrm{nnPU}}(v; P', U') = \begin{cases} 0, & v^* > 1, \\ 4(W_{\mathsf{p}} + W_{\mathsf{n}})v^*(1-v^*), & \text{otherwise.} \end{cases}$$

For the logistic loss we shall again start by computing the derivative of the partial empirical risk:

$$\widehat{R}_{\mathrm{nnPU}}(v; P', U') = W_{\mathsf{p}} \ln(1 + e^{-v}) + \max\{0, W_{\mathsf{n}} \ln(1 + e^v)\}$$

$$\frac{\mathrm{d}\widehat{R}_{\mathrm{nnPU}}}{\mathrm{d}v} = \frac{\max\{0, W_{\mathsf{n}}\}e^v - W_{\mathsf{p}}}{1 + e^v}.$$

In this case, the minimizer $\operatorname{argmin}_{v \in \mathbb{R}} \widehat{R}_{\mathrm{nnPU}}(v; P', U')$ is the same as in (13). The corresponding minimum partial risk is

$$\widehat{R}^*_{\mathrm{nnPU}}(P', U') = \begin{cases} (W_{\mathsf{p}} + W_{\mathsf{n}})(-v^* \ln(v^*) - (1 - v^*) \ln(1 - v^*)), & 0 < v^* < 1, \\ 0, & \text{otherwise.} \end{cases}$$

$\square$

# E   Optimizing Split Time Complexity

Algorithm 2 gives pseudocode for finding the optimal split for a given feature. Each step is annotated with its time complexity, where $n = |P'| + |U'|$. Note that computing $\widehat{R}^*$ value can be done in constant time if $W_{\mathsf{p}}$ and $W_{\mathsf{n}}$ values are given. The time complexity of Algorithm 2 is $O((|P'| + |U'|) \ln(|P'| + |U'|) + m)$, with $m \leq |P'| + |U'| + 1$.

# F   Proof of Proposition 3

In addition, we show that the sigmoid loss, $\widehat{R}^*_{\mathrm{uPU}}(P', U')$ has an interesting interpretation.

**Proposition 3.** *The $\widehat{R}^*(S)$ value is the same for the quadratic loss and the savage loss.*

*Proof.* We have already shown that for the quadratic loss, $\widehat{R}^*(S) = 2|S|wG(S) = 4|S|wq_+q_-$.

For the savage loss, we have

$$\widehat{R}(v; S) = \sum_{(x,y)} w \frac{4}{(1 + e^{vy})^2} = |S|w\left[q_+ \frac{4}{(1 + e^v)^2} + q_- \frac{4}{(1 + e^{-v})^2}\right].$$

---
**Algorithm 2:** `Find Optimal Threshold`

---

**Input:** Feature $f$, set $P'$ of positive examples and set $U'$ of unlabeled examples.

**Output:** Split $(f, t)$ that gives the largest risk reduction.

1   Sort all examples $\boldsymbol{x}_1, \ldots, \boldsymbol{x}_n$ in $P' \cup U'$ by their entries in feature $f$. Denote the sorted
   examples by $\boldsymbol{x}_{(1)}, \ldots, \boldsymbol{x}_{(n)}$ ;                              # $O(n \ln n)$

2   Let the distinct $f$ values be $u_1 < u_2 < \ldots < u_{m+1}$, and choose $t_i = (u_i + u_{i+1})/2$,
   $i = 1, \ldots, m$. Note $m + 1 \le n$ ;                                     # $O(m)$

3   Compute the $\widehat{R}^*(P', U')$ value by computing its $W_{\mathsf{p}}$ and $W_{\mathsf{n}}$ values ;        # $O(n)$

4   Compute the risk reduction for $(f, t_1)$ by computing the $W_{\mathsf{p}}$ and $W_{\mathsf{n}}$ values for the child nodes ;
   # $O(n)$

5   $t^* \leftarrow t_1$ ;                                                         # $O(1)$

6   **for** $i = 2, \ldots, m$ **do** // $O(n)$ `in total`

7      Compute the risk reduction for $(f, t_i)$. This only requires updating $W_{\mathsf{p}}$ and $W_{\mathsf{n}}$ values for
       $t_{i-1}$ by considering those $\boldsymbol{x}_{(i)}$ with $f$ values between $t_{i-1}$ and $t_i$;

8      **if** *Risk reduction for* $(f, t_i) >$ *risk reduction for* $(f, t^*)$ **then**

9        $t^* \leftarrow t_i$;

10     **end**

11 **end**

12 **Return**: $t^*$.

---

The derivative of $\widehat{R}(v; S)$ wrt $v$ is

$$\frac{\mathrm{d}\widehat{R}}{\mathrm{d}v} = 8|S|w\frac{-q_+e^v + q_-e^{2v}}{(1+e^v)^3}.$$

Setting the derivative to 0, we obtain $v_S^* = \ln\frac{q_+}{q_-}$. Therefore

$$\widehat{R}^*(S) = |S|w\left[q_+\frac{4}{(1+q_+/q_-)^2} + q_-\frac{4}{(1+q_-/q_+)^2}\right] = 4|S|wq_+q_-,$$

which is the same as the $\widehat{R}^*(S)$ value for the quadratic loss.          $\square$

**Proposition 5.** *For the sigmoid loss, we have*

$$\widehat{R}^*_{\mathrm{uPU}}(P', U') = \min\{W_{\mathsf{p}}, W_{\mathsf{n}}\}. \tag{16}$$

Intuitively, the above expression uses the weight of the rarer class as the impurity measure, and we can derive a similar expression in the PN setting too. Such an impurity measure is an interesting one that has not been used in decision tree learning.

*Proof.* First note that

$$\widehat{R}_{\mathrm{uPU}}(v; P', U') = \sum_{\boldsymbol{x}\in P'}\frac{w_{\mathsf{p}}}{1+e^v} - \sum_{\boldsymbol{x}\in P'}\frac{w_{\mathsf{p}}}{1+e^{-v}} + \sum_{\boldsymbol{x}\in U'}\frac{w_{\mathsf{u}}}{1+e^{-v}}$$

$$= \frac{W_{\mathsf{p}}}{1+e^v} + \frac{W_{\mathsf{n}}}{1+e^{-v}}$$

The derivative of the partial empirical risk is

$$\frac{\mathrm{d}\widehat{R}_{\mathrm{uPU}}}{\mathrm{d}v} = \frac{e^v(W_{\mathsf{n}} - W_{\mathsf{p}})}{(1+e^v)^2}. \tag{17}$$

The equation (17) tells us that there are no stationary points in the partial empirical risk. The minimizer $v_{P', U'}^*$ is given by:

$$v_{P', U'}^* = \begin{cases} +\infty, & W_{\mathsf{p}} > W_{\mathsf{n}}, \\ \text{any value}, & W_{\mathsf{p}} = W_{\mathsf{n}}, = \\ -\infty, & W_{\mathsf{p}} < W_{\mathsf{n}}, \end{cases} \begin{cases} +\infty, & v^* > 0.5, \\ \text{any value}, & v^* = 0.5, \\ -\infty, & v^* < 0.5, \end{cases}$$

The corresponding minimum partial empirical risk is given by

$$\widehat{R}_{\mathrm{uPU}}^*(P', U') = \begin{cases} W_{\mathsf{n}}, & v^* > 0.5, \\ W_{\mathsf{p}}, & v^* \leqslant 0.5. \end{cases} = \min\{W_{\mathsf{p}}, W_{\mathsf{n}}\}. \tag{18}$$

$\square$

Note that we can derive similar expressions for each loss function described in Section 2, however in this paper we only focus on the quadratic and logistic loss due to their connection with the traditional Gini and entropy impurities, respectively.

## G Proof of Proposition 4

**Proposition 4.** *If a node contains the examples $P' \subseteq P$ and $U' \subseteq U$, $v^*$ is as defined in Proposition 1, and the prediction is chosen from $\{-1, +1\}$ to minimize the empirical risk estimate on $P'$ and $U'$, then the optimal prediction is $2\mathbb{1}(v^* > 0.5) - 1$ for both the* uPU *and* nnPU *risk estimate and all the loss functions in Section 2. Intuitively, we predict $+1$ iff the estimated proportion of positive data at the current node is larger than 0.5.*

*Proof.* For each loss function mentioned in Section 2 we have $\ell(v, y) = a$ if $v = y$ and $\ell(v, y) = b$ if $v \neq y$, for some $0 \leq a < b$ (with $a, b$ not necessarily common to all loss functions). The optimal prediction based on the uPU risk estimator is $+1$ if $\widehat{R}_{\mathrm{uPU}}(+1; P', U') < \widehat{R}_{\mathrm{uPU}}(-1; P', U')$, that is, if

$$W_{\mathsf{p}}\ell(+1, +1) + W_{\mathsf{n}}\ell(+1, -1) < W_{\mathsf{p}}\ell(-1, +1) + W_{\mathsf{n}}\ell(-1, -1)$$
$$W_{\mathsf{p}}a + W_{\mathsf{n}}b < W_{\mathsf{p}}b + W_{\mathsf{n}}a$$
$$(W_{\mathsf{p}} + W_{\mathsf{n}})/2 < W_{\mathsf{p}},$$

which happens if $v^* > 0.5$. A symmetric argument can be applied to show that the optimal prediction is $-1$ when $v^* < 0.5$. Continuing, observe that $a, b \geq 0$, so for the nnPU risk estimator the optimal prediction is $+1$ if

$$W_{\mathsf{p}}a + \max\{0, W_{\mathsf{n}}b\} < W_{\mathsf{p}}b + \max\{0, W_{\mathsf{n}}a\}$$
$$(b - a)\max\{0, W_{\mathsf{n}}\} < W_{\mathsf{p}}(b - a)$$
$$W_{\mathsf{n}} \leq \max\{0, W_{\mathsf{n}}\} < W_{\mathsf{p}}$$
$$W_{\mathsf{n}} < W_{\mathsf{p}},$$

which is the same result as for the uPU risk estimator. A symmetric argument shows that the optimal constant prediction based on the nnPU risk estimator is $-1$ if $v^* < 0.5$, and hence (if we predict $-1$ when $v^* = 0.5$),

$$2\mathbb{1}(v^* > 0.5) - 1 = \mathrm{argmin}_{v \in \{-1, +1\}}\, \widehat{R}_{\mathrm{nnPU}}(v; P', U').$$

We stress that this result holds for any loss function $\ell$ satisfying $\ell(v, y) > \ell(y, y) \geq 0$, with $v \neq y$. Many common loss functions satisfy this condition, including all loss functions mentioned in Section 2 as well as the hinge loss $\ell(v, y) = \max\{0, 1 - vy\}$, the double hinge loss $\ell(v, y) = \max\{0, (1 - vy)/2, -vy\}$, the zero-one loss $\ell(v, y) = (1 - \mathrm{sign}(vy))/2$, the ramp loss $\ell(v, y) = \max\{0, \min\{1, (1 - vy)/2\}\}$ and the exponential loss $\ell(v, y) = \exp(-vy)$. $\square$

## H Pseudocode for PU ET

Our PU Extra Trees algorithm is shown in Algorithm 3, with the learning algorithm for each individual decision tree given in Algorithm 4, and the method used to find a split for a node in Algorithm 5. A nice property of this algorithm is that each tree in the forest can be trained in parallel with no need to share data between processes. Notice the similarity between Algorithm 4 and decision tree learning in the PN setting seen in Algorithm 1, with the differences being our new method for computing the prediction value at a specific node and the sub-routine for choosing a split for a node as in Algorithm 5.

To construct a random forest with 100 decision trees using the randomisation trick in Extra Trees [13], one can run $\mathrm{PUET}(100, P, U, \lceil\sqrt{d}\rceil, 1)$.

**Algorithm 3:** PUET($N, P, U, F, T$)

---

**Input:** $N$ - number of trees; $P$ - positive examples; $U$ - unlabeled examples; $F$ - number of attributes; $T$ - number of threshold values per attribute.

**Output:** Collection of trained decision trees.

1   Initialise collection of decision trees $\mathcal{T} \leftarrow \emptyset$;
2   **for** $i = 1, \ldots, N$ **do**
3      Initialise the root node $\kappa_i$ for decision tree $i$;
4      Train a single decision tree $\mathbb{T}$ using `Construct_Subtree`($\kappa_i, P, U, F, T$);
5      $\mathcal{T} \leftarrow \mathcal{T} \cup \mathbb{T}$;
6   **end**
7   **Return**: $\mathcal{T}$

---

**Algorithm 4:** Construct_Subtree($\kappa, P', U', F, T$)

---

**Input:** $\kappa$ - node; $P'$ - positive examples at $\kappa$; $U'$ - unlabeled examples at $\kappa$; $F$ -number of attributes; $T$ - number of threshold values per attribute.

**Output:** A (sub) decision tree $\mathbb{T}_\kappa$.

1   **if** *termination criterion is met* **then**
2      Compute optimal prediction value $v^*_{P',U'} = \mathrm{argmin}_{v \in \{-1,+1\}} \widehat{R}_{\mathrm{nnPU}}(v; P', U')$ for node $\kappa$;
3   **else**
4      Choose a split $(f, t)$ for node $\kappa$ using `Find_Split`($\kappa, F, T$);
5      Create two child nodes $\kappa_{f>t}$ and $\kappa_{f\leq t}$ for $\kappa$ and split data into $P'_{f>t}, P'_{f\leq t}, U'_{f>t}, U'_{f\leq t}$;
6      $\mathbb{T}_{\kappa_{f>t}} \leftarrow$ `Construct_Subtree`($\kappa_{f>t}, P'_{f>t}, U'_{f>t}, F, T$);
7      $\mathbb{T}_{\kappa_{f\leq t}} \leftarrow$ `Construct_Subtree`($\kappa_{f\leq t}, P'_{f\leq t}, U'_{f\leq t}, F, T$);
8   **end**
9   **Return**: $\mathbb{T}_\kappa$

---

**Algorithm 5:** Find_Split($\kappa, F, T$)

---

**Input:** $\kappa$ - node; $F$ - number of attributes; $T$ - number of threshold values per attribute.

**Output:** Optimal split $(f, t)$.

1   Initialise collection of splits $\mathcal{S} \leftarrow \emptyset$;
2   Select $F$ attributes $\{f\}$ uniformly at random from all non-constant attributes at node $\kappa$;
3   **for** *each* $\{f\}$ **do**
4      **for** $i = 1, \ldots, T$ **do**
5          Sample a random cut-point $t$ uniformly from the range of attribute $f$ at node $\kappa$;
6          $\mathcal{S} \leftarrow \mathcal{S} \cup \{(f, t)\}$;
7      **end**
8   **end**
9   Choose splitting point $(f_\kappa, t_\kappa) \in \mathcal{S}$ that maximises the PU-data based risk reduction
     $\widehat{R}^*(P', U') - \widehat{R}^*(P'_{f>t}, U'_{f>t}) - \widehat{R}^*(P'_{f\leq t}, U'_{f\leq t})$ for node $k$;
10   **Return**: $(f, t)$.

---

# I  Computational Efficiency of PU ET

We have run additional experiments to support our claim that PU ET is efficient. Our current Python implementation of PU ET (nnPU risk with quadratic loss) takes seconds to train a single-tree random forest on modest hardware (Intel Core i7-10700 CPU and 32 GB RAM) as shown in Table 8. Thus training a random forest is typically must faster than training a neural net.

Table 8: Training time mean (sd) in seconds for a single tree random forest.

| Dataset | PU ET | PN ET (ours) | PN ET (sklearn) |
|---|---|---|---|
| 20news | 1.26 (0.03) | 5.39 (0.09) | 0.03 (0.01) |
| Mushroom | 0.14 (0.02) | 0.07 (0.01) | 0.01 (0.01) |
| MNIST | 5.90 (0.33) | 37.90 (0.89) | 0.34 (0.01) |
| CIFAR-10 | 16.32 (0.36) | 119.78 (1.58) | 0.97 (0.01) |
| UNSW-NB15 | 3.35 (0.38) | 67.60 (1.48) | 0.11 (0.01) |

A Cython implementation of PU ET will be much faster: sklearn's cython based implementation of PN ET (with Gini impurity) often achieves a speedup of two orders of magnitude as compared to our PN ET (with Gini impurity), on fully labeled datasets, as shown in Table 8. Note that our PN ET implementation is the same as our PU ET implementation, except different risk measures are used in the calculation of the splitting points.

# J  Hyperparameter Tuning for PU ET

We deliberately chose to not perform hyperparameter tuning for results in the main text to demonstrate that PU ET can be reasonably effective even without significant hyperparameter tuning. We have run some additional experiments to show that the default choice of hyperparameters offer strong performance with the benefit of decreased training times. We trained PU ET with nnPU risk estimator and quadratic loss by varying $F$ (number of sampled features) and $T$ (number of thresholds sampled for each feature) to see their effect on training time and predictive performance. Results are given in Table 9 and Table 10. All experiments here were performed on an Intel Core i7-10700 CPU with 32 GB RAM using MNIST digits, 5 replications, 100 trees in the forest and under the usual PU learning experimental setup ($|P| = 1000, |U| = n$).

Table 9: Accuracy mean% (sd) on the test set for PU ET using various hyperparameter combinations.

| $F$ | $T = 10$ | $T = 1$ |
|---|---|---|
| $d$ | 89.75 (0.71) | 92.47 (0.54) |
| $\lceil \sqrt{d} \rceil$ | 93.24 (0.28) | 93.74 (0.75) |
| 1 | 90.02 (0.68) | 81.03 (0.09) |

Table 10: Training time mean (sd) in seconds for PU ET using various hyperparameter combinations.

| $F$ | $T = 10$ | $T = 1$ |
|---|---|---|
| $d$ | 15071.88 (172.07) | 1771.92 (31.24) |
| $\lceil \sqrt{d} \rceil$ | 1231.89 (26.64) | 225.65 (1.65) |
| 1 | 370.54 (1.22) | 315.42 (5.58) |

The default setting ($F = \lceil \sqrt{d} \rceil, T = 1$) results in lower training times and relatively strong predictive performance. This matches the findings in the original ET paper [13] where this default was able to strike a good balance between training time and predictive performance.

# K  Overfitting with uPU Risk Estimators in PU ET

We performed additional experiments to empirically investigate the difference between uPU and nnPU risk estimators in regards to overfitting. In Table 11 we report the training risks (measured

as PU risk as data is PU) and testing risks (measured as PN risk as data is PN) using zero-one loss $\ell_{0/1}(v, y) = (1 - \text{sign}(vy))/2$ on a number of datasets. Each result is reported as mean (sd) over 5 replications. From the results we can see that the training risk is significantly smaller than the test risk in the uPU setting as compared to the nnPU setting, confirming that uPU suffers more from overfitting than nnPU.

Table 11: Training and testing risk of PU ET.

| Dataset | | uPU | | nnPU | |
| --- | --- | --- | --- | --- | --- |
| | | Quadratic | Logistic | Quadratic | Logistic |
| 20News | Train | -0.48 (0.00) | -0.22 (0.01) | 0.00 (0.00) | 0.00 (0.00) |
| | Test | 0.56 (0.00) | 0.28 (0.01) | 0.15 (0.00) | 0.18 (0.01) |
| Mushroom | Train | -0.36 (0.00) | -0.01 (0.00) | 0.00 (0.00) | 0.00 (0.00) |
| | Test | 0.39 (0.01) | 0.01 (0.00) | 0.00 (0.00) | 0.00 (0.00) |
| MNIST | Train | -0.47 (0.00) | -0.08 (0.01) | 0.00 (0.00) | 0.00 (0.00) |
| | Test | 0.49 (0.00) | 0.11 (0.01) | 0.04 (0.00) | 0.06 (0.00) |
| CIFAR-10 | Train | -0.38 (0.00) | -0.17 (0.01) | 0.00 (0.00) | 0.00 (0.00) |
| | Test | 0.4 (0.00) | 0.25 (0.00) | 0.16 (0.00) | 0.21 (0.00) |
| UNSW-NB15 | Train | -0.58 (0.00) | -0.03 (0.01) | 0.02 (0.01) | 0.02 (0.01) |
| | Test | 0.65 (0.00) | 0.13 (0.01) | 0.1 (0.01) | 0.14 (0.00) |

## L    MNIST Feature Importances

For MNIST, the plot of the importance scores for PU ET for each digit often suggests the shape of the digit. Figure 4 shows that the normalized risk reduction importance makes many more pixels more important. This observation is consistent with our discussion in Section 4. Interestingly, the importances for PU ET appear to be very similar to those for PN ET using both the risk reduction importance and normalised risk reduction importance. This suggests that the learnt PU model is likely quite similar to the learnt PN model.

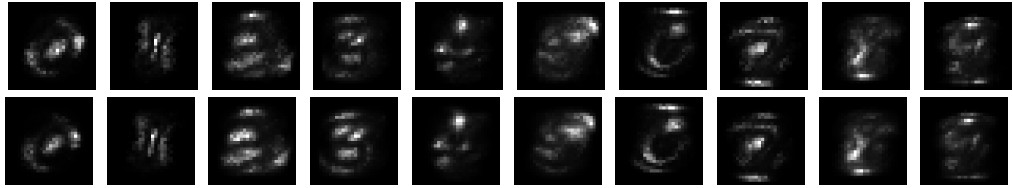

Figure 3: Risk reduction importance scores for MNIST digits. Top: PU ET with nonnegative risk estimator and quadratic loss. Bottom: PN ET with quadratic loss.

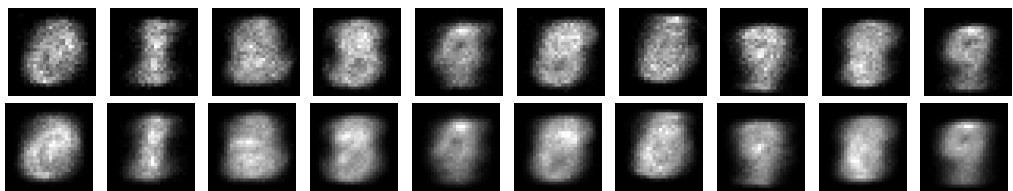

Figure 4: Normalized risk (impurity) reduction importances for MNIST digits. Top: PU ET with nonnegative risk estimator and quadratic loss. Bottom: PN ET with quadratic loss.

## M    Feature selection with risk reduction importance

We demonstrate that the risk reduction importance is more capable of selecting features which are responsible for risk minimization and the generalization performance. We do this on the UNSW-NB15 dataset. Figure 5 shows how the accuracy of PU ET changes on UNSW-NB15 when using only top features selected by risk reduction importance and Gini impurity reduction importance. All

accuracies are reported as the mean over five replications to account for randomness in sampling the P data used during training and randomness in the decision tree construction process.

With the same number of selected features, the risk reduction importance generally leads to increased accuracy on the test data compared to using Gini impurity reduction on the UNSW-NB15 dataset. The difference in area under the curve is 0.84 in favour of the risk reduction importance. This confirms that the risk reduction importance is indeed more useful for selecting features which are responsible for risk minimization and the generalization performance.

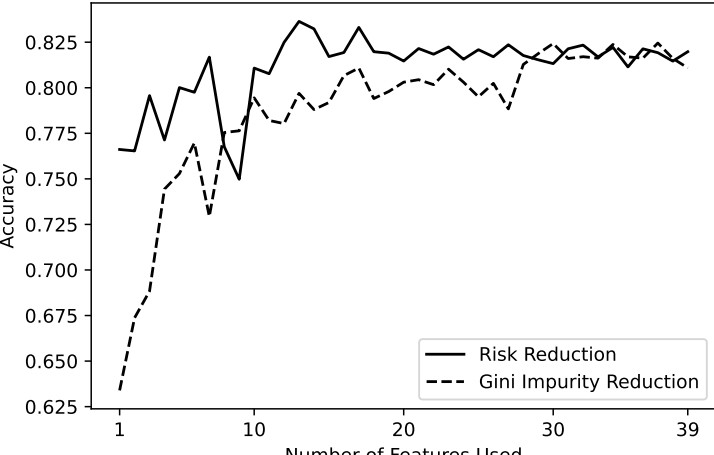

Figure 5: Risk reduction importance and Gini impurity reduction importance when using different numbers of features.