# OpenReview forum: "Positive-Unlabeled Learning using Random Forests via Recursive Greedy Risk Minimization"
_NeurIPS.cc/2022/Conference — NeurIPS 2022 Accept_

### Official Review · Reviewer_J3hT · 2022-07-10

**Rating:** 6
**Confidence:** 4
**Soundness:** 2 fair
**Presentation:** 2 fair
**Contribution:** 3 good

**Summary:**

This paper proposes new random forest algorithms for PU-learning and further gives a new interpretation of decision tree algorithms for positive and negative data as recursive greedy risk minimization algorithms. Specifically, the authors extend the popular risk estimator on PU data to split criteria of tree-based methods. Conclusively, it is an interesting trial for both PU-learning and tree-based methods.

AFTER REBUTTAL:  Authors have solved most of my concerns. I plan to increase my score and suggest acceptance of this work.

**Questions:**

Questions/suggestions
1. Could authors provide more detailed experimental settings to verify the results. It’s acceptable that a tree-based method could not outperform deep neural network with lower time cost or better interpretability.
2. Authors may add the missing computational time details and a comparison of computational considerations between the proposed PU ET and recent PU learning methods. I am more concerned of actual time consumption compared with deep neural networks.
3. More settings in PU learning may be explored.
4. Could authors provide a better illustration of interpretability other than feature importance, maybe from the perspective of the decision-making path.
5. In the Introduction (Section 1), the authors claim that “Existing tree-based methods have not been designed to directly minimize the loss.”. Combined with the propositions in Recursive Greedy Risk Minimization (Section 3) such declaration and theorem seem a bit hand-wavy since the nature of a decision tree could be perceived as utilizing different loss functions that evaluate the split based on the purity of the resulting nodes.


**Limitations:**

Yes

**Strengths And Weaknesses:**

Strengths
1. The motivation of this paper is strong. The extension of popular risk estimator on PU data to the tree-based methods is also very interesting at least for me.
2. This paper provides complete theories and is easy to follow.

Weaknesses
1. Several promising properties of tree-based methods like fastness and interpretability are not fully illustrated in the paper. No significant advantages are showed compared with existing PU methods. Especially for the perspective of interpretability, only feature importance is not convincing enough considering similar works could be also be done by neural networks.
2. The reported performances of uPU, nnPU and Self-PU in this paper have a large gap compared with other recent papers [1][2][3] and this makes me confused. Could authors provide more detailed experimental settings to verify the results?
3. Several important data settings in PU learning like data imbalance are not mentioned. Plus, does the proposed approach rely on the "selected completely at random" (SCAR) assumption?

[1] Li, Changchun, et al. "Who Is Your Right Mixup Partner in Positive and Unlabeled Learning." International Conference on Learning Representations. 2021.
[2] Chen, Hui, et al. "A variational approach for learning from positive and unlabeled data." Advances in Neural Information Processing Systems 33 (2020): 14844-14854.
[3] Chen, Xuxi, et al. "Self-pu: Self boosted and calibrated positive-unlabeled training. "International Conference on Machine Learning. PMLR, 2020.

---

> ### Author Response · Authors · 2022-08-06
> **Response to Reviewer J3hT Part 1/2**
>
> Thank you for your helpful feedback and suggestions!
>
> **Advantage as compared to existing PU methods**
>
> We respectfully disagree that PU ET demonstrate no significant advantage as compared with existing PU methods. Importantly, PU ET achieves competitive accuracies and F-scores on multiple problems as compared to existing neural network based methods, at the same time requires little hyperparameter tuning and supports a feature importance that directly measures a feature's contribution to risk minimization. We have also run additional experiments to compare with existing tree-based methods, and PU ET shows strong performance. Please kindly refer to *Empirical comparison with other tree-based methods* in our response to Reviewer 4GSs.
>
> **Empirical support for the computational efficiency of PU-ET**
>
> We have performed experiments to demonstrate the computational efficiency of PU ET. Please kindly refer to our response to Reviewer jd1V.
>
> **Gap between our and other reported performances for uPU, nnPU and Self-PU (on MNIST and CIFAR-10)**
>
> Thank you for the references. Our experiments were carefully designed to ensure that results from the original papers that introduced the methods included in our comparison were reproduced as closely as possible. We have also spent a lot of time to investigate the differences between the results (checking our and other available code, rerunning some experiments, reading the papers closely). Currently we have noted the following differences between the evaluations.
> * [a] (CIFAR-10 only) Their uPU and nnPU architectures are different from the nnPU paper [15] (which we followed), and its Self-PU result is lower than ours.
> * [b] (CIFAR-10 only) They used 3000 positive examples, while we and other referenced papers used 1000. We used the architecture in [15], while they used a different architecture.
> * [c] (MNIST and CIFAR-10) We used the PU split in [15], while they used a different MNIST PU split so the results are not directly comparable. For CIFAR-10, they reported best accuracy over 200 epochs, while we report accuracy after 200 epochs.
>
> In addition, some details (e.g., complete architecture specification) seem to be missing in some papers, and this makes it harder to compare the results. We leave a systematic and thorough evaluation of existing methods to future work.
>
> * [a]. Li, Changchun, et al. "Who Is Your Right Mixup Partner in Positive and Unlabeled Learning." International
> Conference on Learning Representations. 2021.
> * [b] Chen, Hui, et al. "A variational approach for learning from positive and unlabeled data." Advances in Neural Information Processing Systems 33 (2020): 14844-14854.
> * [c] Chen, Xuxi, et al. "Self-pu: Self boosted and calibrated positive-unlabeled training. "International Conference on Machine Learning. PMLR, 2020.

---

> ### Author Response · Authors · 2022-08-06
> **Response to Reviewer J3hT Part 2/2**
>
> **Responses to other comments**
> * Detailed experimental settings: The experiment settings are provided in lines 241-245 and we have made our source code available at https://github.com/puetpaper/PUExtraTrees for reproducibility. We will include the following missing details:
>     * GloVe pre-trained word embeddings were used for 20News with average pooling over each document.
>     * A 6 layer MLP with ReLU was used for MNIST, Covtype, Mushroom and UNSW-NB15; a similar model was used for epsilon while the activation was replaced with Softsign; a 5 layer MLP with Softsign was used for 20News. All hidden layers had width 300 for the MLPs. The model for CIFAR-10 was the 13 layer CNN: (32 * 32 * 3,1)-[C(3 * 3,96,1)] * 2-C(3 * 3,96,2)-[C(3 * 3,192,1)] * 2-C(3 * 3,192,2)-C(3 * 3,192,1)-C(1 * 1,192,1)-C(1 * 1,10,1)-1000-1000-1, where the input is a 32 * 32 RGB image, C(3 * 3,96,1) means 96 channels of 3 * 3 convolutions with stride 1 followed by ReLU, [.]*2 means there are two such layers.
>     * For hyperparameter tuning we performed a grid search for learning rate in {1e-2, 1e-3, 1e-4, 1e-5}, number of mini-batches in {10, 100}, batch norm momentum in {None, 0.1, 0.9} (for nnPU only), optimiser in {adam, adagrad}, weight decay in {None, 1e-4, 5e-8}. The hyperparameters found for nnPU were also used for uPU.
> * Additional PU learning settings: We focused on papers most closely related to our work. We will highlight that many alternative settings have been considered by directing readers to the survey [12] cited in our paper.
> * SCAR assumption: Our data generation assumption (line 82-85) is different from the SCAR assumption.
> * Illustration of interpretability other than feature importance, from the perspective of decision paths: Our emphasis is that PU ET supports a feature importance that directly measures a feature's contribution to risk minimization, and we believe this is important on its own. Besides the interesting feature importance heatmaps for MNIST, we also illustrated in Fig. 5 in the supplementary file that our feature importance scores can effectively select useful features. We appreciate the suggestion of further illustration of interpretability from the perspective of decision paths. This is often done for a single decision tree, and we will investigate this for forests in future work.
> * The claim that “Existing tree-based methods have not been designed to directly minimize the loss.” and our theory are "hand-wavy": We believe we have presented our ideas clearly and rigorously (Reviewer 4GSs considered our mathematical development sound). We are not sure why the reviewer consider our claim and results "hand-wavy". We'll be happy to provide further explanation if the reviewer could clarify this.

---

> > ### Comment · Reviewer_J3hT · 2022-08-06
> > **Response to “Existing tree-based methods have not been designed to directly minimize the loss.”**
> >
> > (a) I appreciate the implementation details and extra experiments given by authors. Some explanations have resolved my confusions (The gap with other papers and time consumption). But my questions on interpretability still holds.
> > (b) In response to authors' confusion, specifically, the proposed algorithm resembles the popular Extra-Tree algorithm and a modified split criterion. However, in my opinion, most existing tree-based methods can be simply perceived as a recursive process of optimizing local metrics (losses) like information gain or Gini index. It is new to implement tree-based method on PU but it is not convincing to me such straightforward modification is novel enough especially considering the risk(uPU,nnPU) is widely used.
> > (c) In the respect of theorem, I think theorem 1 were already widely known and the illustration is solid in previous work. Other minor mistakes also exist in propositions. (I just noticed that the reviewer 4GSs points out most of my concerns and the response to 4GSs solve these concerns).

---

> > > ### Author Response · Authors · 2022-08-07
> > > **response**
> > >
> > > Thank you for confirming that we have addressed some of your comments!
> > >
> > > a. Our claimed contribution is a feature importance that directly measures a feature’s contribution to risk minimization. To the best of our knowledge, this is novel in the PU case, and it provides a nice generalization of the Gini importance in the PN case. We illustrated that our PU feature importance can be similar to its PN counterpart despite using a much less informative dataset (Section 5.2), can effectively inform us about the important features are important (Section 5.2), and can select a small subset of useful features (Fig. 5 in the supplement). This contributes to the interpretability of the algorithm. Overall, while we agree that it'll be helpful to develop more interpretability mechanisms, we believe our feature importance is a novel and important contribution. If you've any suggestion on what else can possibly be done, please kindly let us know and we would very much appreciate that!
> > >
> > > b. While our algorithm is indeed similar to PN tree algorithms "optimizing *local* metrics (losses) like information gain or Gini index", we would like to clarify that we derive this in a principled way by minimizing the *global* empirical risk on the entire dataset, and our idea supports a random forest variant too (Section 4). We would also like to highlight that, importantly, while we have consciously made an effort to expound the analogy between the PN and PU case, our PU tree learning algorithm involve several subtle/counterintuitive aspects not present in the PN case (See *Range of the unbiased probability estimates $W_{\mathrm{p}}$ and $W_{\mathrm{n}}$* in response to Reviewer jd1V, *Why quadratic loss is much poorer than logistic loss for PU ET optimizing uPU risk* in reponse to 4GSs. We also note that while our prediction rule in Proposition 4 is presented in a simple form, this is not an obvious result either.). In addition, the fact that uPU/nnPU risk is widely used but no tree-based algorithm has been developed shows that our work fills in an important gap, particularly considering that our resulting algorithm is robust to hyperparameter choice and naturally offers a feature importance measure that directly measures how much a feature contributes to risk minimization.
> > >
> > > c. We will add the reference pointed out by Reviewer 4GSs to Theorem 1(a) to the final version of the paper, and we have fixed errors in the submitted supplements. We're not sure about "theorem 1 were already widely known and the illustration is solid in previous work". Does it mean Theorem 1(b) appears in some previous work too? We'd be happy to add the reference in that case!

---

> > > ### Author Response · Authors · 2022-08-09
> > > **Further feedback**
> > >
> > > Dear Reviewer J3hT, thank you for your helpful comments and discussion so far. We believe we've addressed your remaining concerns on the novelty and significance of our work by highlighting the non-trivial aspects of our work, the importance of filling in a gap of effective PU tree-based methods, the usefulness of our novel feature importance measure, with support from our additional explanations and experimental support for these claims. If there are any remaining or further concerns, we'd like to take the opportunity to address them before the discussion period closes soon. If our response has resolved your concerns, we'd appreciate you increasing your score. Thank you!

---

### Official Review · Reviewer_Jd1V · 2022-07-10

**Rating:** 6
**Confidence:** 3
**Soundness:** 3 good
**Presentation:** 3 good
**Contribution:** 3 good

**Summary:**

In the present paper, the authors focus on the problem of PU learning, a peculiar ML setting where the dataset of a binary classification problem only contains positive and unlabeled samples (no negative ones). The main goal is to derive an algorithm for training ensembles of decision trees in this setting (the authors choose the ExtraTrees as a simple and efficient solution), and the required step is a connection between the usual impurity reduction schemes used in normal decision trees and the modified risk minimization problem of the PU setting. Once the ad hoc splitting method is defined, the authors proceed to compare the novel method with some NN baselines. Moreover, they argue that the ExtraT setting allows an interpretation of the learned model through the typical feature importance evaluation, in this case directly connected to the PU risk minimization problem.

AFTER REBUTTAL
The empirical evaluation of the proposed method has been improved in the revised version and most of the concerns raised by the reviewers have been adequately addressed.

**Questions:**

- lines 92-93-94: it is not clear where these alternative losses are employed in this paper. Are these definitions truly necessary?
- in the nnPU risk, the authors propose enforcing a non-negative contribution to the negative examples. This was already proposed in another paper, but it is necessary for a new reader to have some explanation on why this assumption is useful, and when it may fail or cause even more over-fitting than the unbiased estimator.
- 147-151 : these lines are a little confusing as unbiased estimates of probabilities seem not to fall in the correct interval 0<p<1. could the authors clarify this passage?
- 226-233: this paragraph seems to assume that the reader is familiar with the previous work of the authors, but the explanation of the problem setting where the method was assessed should be self-contained.
- 271-275 and figure 2: this result seems very surprising, especially given that the performance in the PU setting cannot be as good as the one in the PN setting. Can the authors expand on the sentence "the PU model is likely quite similar to the PN model" and look at this more thoroughly?

I believe the authors need to find some way to access better computational resources in order to offer better numerical support for their results:
- 220-222: Assessing the computational cost of a novel learning method should be one of the important points of the paper that introduces it. The short comment provided here seems very vague.
- in lines 238-240 the authors state that the standard parameters for extra trees were employed, however, in the tables (e.g. 2-3) the parameters seem to differ. Again, it seems that the computational constraints were too many for the authors to perform their numerical tests thoroughly.
- Self-PU seems to work very well (and possibly better than the ET alalgorithm), but the authors admit that the self-calibration was not implemented due to computational constraints. Without it a proper assessment becomes somewhat hard.
- in lines 286-289, the authors mention other attempts at employing decision trees in the PU setting, but none is tested and compared against the novel method.
-Hyperparameter tuning (through crossvalidation) is often key in the context of ensembles of trees but the authors completely skip it due to the lack of computational resources.

**Limitations:**

The authors do not comment on the possible limitations of this work. Please read above for my comments and perplexities.


**Strengths And Weaknesses:**

Given the simplicity and interpretability of decision trees, and their effectiveness when employed in ensembling methods, the contribution of the authors seems pretty relevant: the definition of the learning step in the PU setting is not completely trivial. Moreover, if the connection between typical impurity reduction and risk minimization was not previously shown in other works (which would be rather surprising), the framework presented by the authors could help employ these methods in diverse settings where the typical learning strategy would not be applicable.

However, this work heavily builds (and often relies) on previous work [10,15], and most of the presented content seems not to be very surprising or complicated. All the results follow straightforwardly from the definition of the uPU and the nnPU risks, which however are not novel. The biggest limitation of this work, in my view, is in the numerical experiments presented by the authors, which did not thoroughly explore the potential of their new method, nor compare it with optimized competitors due to a lack of computational power.

---

> ### Author Response · Authors · 2022-08-06
> **Response to Reviewer Jd1V Part 1/2**
>
> Thank you for your detailed and helpful feedback and suggestions!
>
>
> **Novelty of our work**
>
> Please kindly refer to our response to Reviewer xssD. If there are other additional results we have missed that you think could help substantiate our work we are eager to know your opinion.
>
> **Better empirical support for PU ET**
>
> Thank you for your suggestions. We have provided additional results and clarifications below.
>
> *(A) Computational efficiency of PU ET*
>
> We have run additional experiments to support our claim that PU ET is efficient.
> Our current Python implementation of PU ET (nnPU risk with quadratic loss) takes seconds to train a single-tree random forest on modest hardware (Intel Core i7-10700 CPU and 32 GB RAM) as shown below. Thus training a random forest is typically must faster than training a neural net.
>
> | Dataset | PU ET | PN ET (ours) | PN ET (sklearn) |
> | -- | -- | -- | -- |
> | 20news | 1.26 (0.03) | 5.39 (0.09) | 0.03 (0.01) |
> | Mushroom | 0.14 (0.02) | 0.07 (0.01) | 0.01 (0.01) |
> | MNIST | 5.90 (0.33) | 37.90 (0.89) | 0.34 (0.01) |
> | CIFAR-10 | 16.32 (0.36) | 119.78 (1.58) | 0.97 (0.01) |
> | UNSW-NB15 | 3.35 (0.38) | 67.60 (1.48) | 0.11 (0.01) |
>
> A Cython implementation of PU ET will be much faster: sklearn's cython based implementation of PN ET (with Gini impurity) often achieves a speedup of two orders of magnitude as compared to our PN ET (with Gini impurity), on fully labeled datasets, as shown in the table above. Note that our PN ET implementation is the same as our PU ET implementation, except different risk measures are used in the calculation of the splitting points.
>
> *(B) Standardization of choice of hyperparameters for PU ET*
>
> We did use the same hyperparameters for all PU ET experiments, except for the uPU version on Epsilon and Covtype. Note that we already observed that uPU often overfits and do not recommend uPU in general. We are currently rerunning these experiments using the default hyperparameters and will report the results soon.
>
> *(C) Use of self-calibration in Self-PU*
>
> We have used self-calibration in all our experiments, except leaving it out on CIFAR-10 due to hardware constraint. We have done further investigation to make sure that our comparison remains fair, as follows. Our reported accuracy is 88.22 (0.51)% without self-calibration. The original paper reported 89.68(0.22) without self-calibration and 90.77 (0.22)% with self-calibration. However, we noted that the original paper's code choose best test set accuracy over 200 epochs, while we report the test set accuracy at the last epoch.In addition, our result is better than the accuracy of 85.1% as reported in [a]. Finally, note that PU ET is competitive with or outperforms Self-PU on all tasks except CIFAR-10.
>
> [a] Chen, Hui, et al. "A variational approach for learning from positive and unlabeled data." Advances in Neural Information Processing Systems 33 (2020): 14844-14854.
>
>
> *(D) Comparison with other tree-based methods*
>
> We have provided additional experimental results comparing our methods with other tree-based methods. Please kindly refer to our response to Reviewer 4GSs.
>
> *(E) Tuning hyperparameter for PU ET*
>
> We agree that hyperparameter tuning can potentially further improve the performance of PU ET. We however deliberately chose to not perform hyperparameter tuning to demonstrate that PU ET can be reasonably effective even without significant hyperparameter tuning.
>
> We have run some additional experiments to show that the default choice of hyperparameters offer strong performance with the benefit of decreased training times. We trained PU ET with nnPU risk estimator and quadratic loss by varying $F$ (number of sampled features) and $T$ (number of thresholds sampled for each feature) to see their effect on training time and predictive performance. All experiments here were performed using MNIST digits, 5 replications, 100 trees in the forest and under the usual PU learning experimental setup $(|P| = 1000, |U| = n)$.
>
>
> Accuracies are shown below.
>
> | $F$                    | $T=10$       | $T=1$ |
> | -- | -- | -- |
> | $d$                    | 89.75 (0.71) | 92.47 (0.54) |
> | $\lceil\sqrt{d}\rceil$ | 93.24 (0.28) | 93.74 (0.75) |
> | $1$                    |  90.02 (0.68) | 81.03 (0.09) |
>
>
> Running times are shown below.
>
> | $F$                    | $T=10$         | $T=1$ |
> | -- | -- | -- |
> |  $d$                   | 15071.88 (172.07) | 1771.92 (31.24) |
> | $\lceil\sqrt{d}\rceil$ | 1231.89 (26.64) | 225.65 (1.65) |
> | $1$                    | 370.54 (1.22) | 315.42 (5.58) |
>
> The default setting $(F,T)=(\lceil\sqrt{d} \rceil, 1 )$ results in lower training times and relatively strong predictive performance. This matches the findings in the original ET paper [12] where this default was able to strike a good balance between training time and predictive performance.

---

> > ### Author Response · Authors · 2022-08-08
> > **(B) Standardization of choice of hyperparameters for PU ET**
> >
> > We have now finished rerunning the experiments marked with $^\dagger$ using the same default hyperparameters as in other PU ET experiments. Below are the updated accuracies.
> >
> > | Dataset | uPU Quadratic | uPU Logistic |
> > | -- | -- | -- |
> > | Epsilon | 50.04 (0.00) | 50.67 (0.08) |
> > | Covtype | 48.72 (0.00) | 72.63 (0.87) |
> >
> > Updated F-scores are shown below.
> >
> > | Dataset | uPU Quadratic | uPU Logistic |
> > | -- | -- | -- |
> > | Epsilon | 0.00 (0.00) | 4.18 (0.59) |
> > | Covtype | 0.11 (0.01) | 68.44 (1.25) |
> >
> > In general we see a drop in predictive performance when the restriction on depth is removed, particularly when using the quadratic loss. This suggests that uPU risk estimators are susceptible to overfitting, agreeing with the results of additional experiments investigating overfitting in more detail and the effect of aggressive splitting behaviour caused by the use of uPU risk estimators compared to nnPU. We'll use these results in the final version of the paper.

---

> ### Author Response · Authors · 2022-08-06
> **Response to Reviewer Jd1V Part 2/2**
>
> **Responses to other comments**
> * (92-94) Alternative loss functions: we will remove unused loss functions, and we will make it clear we consider the savage and sigmoid loss in Proposition 3 (corrected in the supplement).
> * More explanation on nnPU risk: We appreciate the suggestion and agree. We will incorporate our discussion on uPU and nnPU in the response to Reviewer 4GSs in the final version of our paper. Regarding when nnPU fails or performs worse than uPU, we agree that it is interesting and we plan to investigate this in future work.
> * (147-151) Range of the unbiased probability estimates $W_{\mathrm{p}}$ and $W_{\mathrm{n}}$: $W_{\mathrm{p}}$ is always in [0, 1], while the estimate $W_{\mathrm{n}}$ is always $\le 1$ and can be negative. $W_{\mathrm{n}}$ is unsual in that it contains a negative compoment. The following derivation will shed insight on this:
> For any $x$, we have $(1-\pi) p_{\mathrm{n}}(x) = (p(x) - \pi p_{\mathrm{p}}(x))$.
> Integrating over $x \in \kappa$, we have
> $(1-\pi) P(y=-1 | x \in \kappa) = P(x \in \kappa) - \pi P(y=1 | x \in \kappa)$,
> or equivalently,
> $P(y=-1, x \in \kappa) = P(x \in \kappa) - \pi P(y=1 | x \in \kappa)$.  Now, $P(x \in R)$ is unbiasedly estimated by $|U'| w_{u}$, and $\pi P(y = 1 | x \in R)$ is unbiasedly estimated by $\pi |P'|/|P| = |P'| w_{\mathrm{p}} = W_{\mathrm{p}}$.
> * (226-233) A more self-contained explanation of the experimental setting: We will add the following missing explanation: MNIST was processed in such a way that even and odd digits formed the P and N classes, for 20News, ‘alt.’, ‘comp.’, ‘misc.’ and ‘rec.’ make up the P class, and ‘sci.’, ‘soc.’ and ‘talk.’ make up the N class; for CIFAR-10, the P class is formed by ‘airplane’, ‘automobile’, ‘ship’ and ‘truck’, and the N class is formed by ‘bird’, ‘cat’, ‘deer’, ‘dog’, ‘frog’ and ‘horse’. The dataset Epsilon comes with two classes and so no such processing is necessary. For 20News, we also borrowed the pre-trained word embeddings from GloVe.
> * (271-275) Similar feature importance scores for PU ET and PN ET: We agree that in general these two types of importance scores are likely to be quite different, as PN ET may be much better than PU ET. In this case (MNIST), PU ET's and PN ET's accuracies are 93% and 98% respectively, thus they may have learned similar information, leading to similar importance scores.

---

> ### Author Response · Authors · 2022-08-08
> **Further feedback**
>
> Dear Reviewer Jd1V, thank you for your time and thoughtful review. We believe we have addressed your concerns in our response, and we'd appreciate you increasing your score if you agree. However, if you've any remaining or further questions for us to address, we're keen to take the opportutnity to do so before the discussion period closes. Thank you!

---

> > ### Comment · Reviewer_Jd1V · 2022-08-08
> > **Response to the rebuttal**
> >
> > I would like to thank the authors for their efforts in explaining and improving the present paper.
> > I think my main concerns were indeed addressed and that the work was solidified through the revision process.
> > For these reasons, I plan to increase my score and suggest acceptance of this work.

---

> > > ### Author Response · Authors · 2022-08-08
> > > **thank you!**
> > >
> > > We very much appreciate your kind support, and thank you again for your time and many helpful comments!

---

### Official Review · Reviewer_xssD · 2022-07-10

**Rating:** 5
**Confidence:** 3
**Soundness:** 2 fair
**Presentation:** 4 excellent
**Contribution:** 2 fair

**Summary:**

This paper proposes a decision tree learning algorithm in PU framework. In this setup only (a subset of) positive class labels are provided, whereas the remaining data is considered unlabeled (it may contain both negative and positive class samples). The algorithm is heavily based on CART-style greedy tree growing procedure where splitting (or purity) criterion is replaced by PU-specific loss. They extend it to train ensemble of trees via Extra Trees framework.

**Questions:**

Please address weaknesses in the previous section and limitations in the section below.

**Limitations:**

I am not sure how the method is extended to multiclass setting. I imagine it will one-vs-one. In this case, computational runtime might be an issue since K trees need to be induced (i.e., one per class).

**Strengths And Weaknesses:**

Strengths:
- Nice and clear presentation;
- Experiments contains important ablation studies, such as comparison of different losses;
- It is nice to see that the method is relatively robust to different factors, such as the choice of loss function;
- Derivation of the empirical risk specifically to quadratic and logistic losses is definitely a valuable contribution.

Weaknesses:
- Novelty. I am not familiar with PU literature but it looks there have been previous works on the topic of PU learning for DTs. Authors claim that the proposed approach is "new DT learning algorithm that directly minimizes PU-data expected risk". However, the algorithm itself resembles CART + "modified purity criterion". By "modified purity criterion", I mean instead of using gini index or cross-entropy (which will be inapplicable in PU setting), authors use PU specific loss which requires a minor modification and straightforward extension to the existing tree growing procedure. On top of that, in order to construct ensemble of trees, authors use well known framework -- ExraTrees.
- Experiments. Authors compare against neural net only which is fine. However, it is critical to have other tree-based baselines given that authors mention previous works on this topic.



---- After Rebuttal --------

Experiments. Afters addressed my concerns regarding comparison with tree-based baselines. Thanks!

Novelty. I still think that the method can be summarized as a modified impurity criterion + regularization. Although there seem to be no literature of such approaches for PU, but semi-supervised learning (SSL) trees use such greedy procedure (see below Levatić et al.).

J. Levatić, M. Ceci, D. Kocev, and S. Džeroski. Semi-supervised classification trees.

Given all these, I still think that the paper lacks a significant level of contribution in terms of novelty. I will keep my original score.

---

> ### Author Response · Authors · 2022-08-06
> **Response to Reviewer xssD**
>
> Thank you for your helpful feedback and suggestions!
>
> **Novelty of our work**
>
> To the best of our knowledge, there is no prior work that presents a general framework for building decision trees to minimize arbitrary risk measure via recursive greedy empirical risk reduction, this is what enabled us to develop our customized PU tree learning algorithm. As Reviewer 4GSs pointed out, we did miss one reference that demonstrates the connection between quadratic loss and Gini impurity as we stated in Theorem 1 (a), however there does not seem to be a similar published result for the logistic loss and entropy impurity; and Reviewer 4GSs considers our PU tree learning algorithm interesting and novel. Furthermore, we can also derive new impurity measures based on other loss functions in the PN setting - for example the sigmoid loss corresponds to an impurity measure that returns the weight of the rarer class at the node (Proposition 5 in the supplement). We believe that this opens up interesting opportunities for further research.
>
>
> As to the resemblance of existing PN tree learning algorithms, we are pleased that our effort to expound the analogy between the PN and PU case has made the idea easy to grasp, but our PU tree learning algorithm involve several subtle/counterintuitive aspects not present in the PN case. See *Range of the unbiased probability estimates $W_{\mathrm{p}}$ and $W_{\mathrm{n}}$* in response to Reviewer jd1V, *Why quadratic loss is much poorer than logistic loss for PU ET optimizing uPU risk* in reponse to 4GSs. We also note that while our prediction rule in Proposition 4 is presented in a simple form, this is not an obvious result either.
>
> Additionally, while we build on the great work of [10,15], these papers do not show how trees can be built to mininize uPU/nnPU risk and how to make predictions with them, and this makes our work differ greatly from these works. We also note that our method is robust to hyperparameter choice and naturally offers a highly interpretable feature importance measure.
>
> If there are other additional results we have missed that you think could help substantiate our work we are eager to know your opinion.
>
> **Empirical comparison with other tree-based baselines**
>
> We appreciate this suggestion, and we have run further experiments for this. Please kindly refer to our response to Reviewer 4GSs for the details.
>
> **Extending PU ET to the multiclass setting**
>
> As in most PU learning papers, we focus on the binary case. We agree that an extension to the multiclass setting is interesting and we plan to investigate this in our future work.

---

> ### Author Response · Authors · 2022-08-08
> **Further feedback**
>
> Dear Reviewer xssD, thank you for your time and thoughtful review. We believe we have addressed your concerns in our response, and we'd appreciate you increasing your score if you agree. However, if you've any remaining or further questions for us to address, we're keen to take the opportutnity to do so before the discussion period closes. Thank you!

---

### Official Review · Reviewer_4GSs · 2022-07-13

**Rating:** 8
**Confidence:** 4
**Soundness:** 4 excellent
**Presentation:** 4 excellent
**Contribution:** 3 good

**Summary:**

This paper adapts the decision tree growing algorithm to handle positive-unlabeled learning, i.e. learning with only positive and unlabeled examples. The authors first note that the standard tree growing algorithm can be interpreted as recursive greedy risk minimization. They thus propose to adapt this idea to grow a tree that minimizes two specific risk estimators for the positive and unlabeled settings, namely uPU and nnPU. For these two risks, they propose a closed-form solution of the minimum risk and the optimum solution in a nodes that can be exploited to efficiently find the best split points at a tree node. Experiments are carried out random forests of such trees on several classification tasks, where the approach is shown to be competitive with neural networks algorithm based on the minimization of similar risk estimates.

**Questions:**

See the previous point.


**Limitations:**

Given the lack of a comparison against other competitors for PU learning with trees or forests, I think the paper does not position well the proposed method in the existing literature. I would suggest to at least add the two simple baselines suggested above in the comparison.

**Strengths And Weaknesses:**

The paper is well written and easy to follow.

The idea explored in the paper is interesting. The adaptation of decision tree splits to minimize the PU risks (uPU and nnPU) is sound and original to the best of my knowledge. The resulting algorithm is as computationally efficient as the standard tree growing algorithm, which is a very nice result. I think the mathematical developments are sound (I quickly checked the supplements and proofs).

Some comments on the method part:

- I would have appreciated some more discussions about uPU and nnPU. I'm not sure why uPU can cause overfitting and how nnPU solves this issue.

- Results in Theorem 1 were already known to me and I thought they were widely known. In the case of quadratic loss and GINI entropy, this is shown in the CART book (see Sections 4.6.2 and 11.2):
Breiman et al., Classification and regression trees, Taylor and Francis, 1984.
In the case of logistic loss and entropy however, I could not find a reference.

- I noted the few mistakes in the propositions that were corrected in the supplement. They need to be corrected in the paper obviously. I think that you should make it clear in the paper that $v^*_{P',u'}$ is not equal to what is denoted by $v^*$ in the case of logistic loss also. The notation $v^*$ is a bit confusing. It should be clear that the exact form of $v^*_{P',u'}$ is only given in the supplement in this case.

- I'm a bit puzzled by the fact that the risk can be equal to $-\infty$ in the case of uPU as soon as $v^*$ becomes higher than 1. Does it mean that as soon as $v^*$ in a given node is greater than 1, one can not split a node anymore (because the risk can not be reduced further)? Since this can happen when there are still positive and unlabeled example in a node, this is a bit counterintuitive to me. Overall, I think that more discussion is needed to explain the consequence on tree growing of using these two risks uPU and nnPU. When do we stop splitting a branch for example?

- About PU ET, the authors mentioned that they implemented a more general version which allows sampling multiple random thresholds for sampled feature. This is not what is done in the pseudocode in the appendix and this version does not seem to be tested in the experiment.

- The discussion about risk reduction importance is confusing. I think the exact formulas used to compute feature importances should be provided because what the authors suggest in their last sentence as a generalization of Gini importance (i.e., summing the risk reduction divided by the total weight of the examples) does not correspond to what is done in Gini importance. In the CART book, the Gini importance of a feature (or more generally the mean decrease of impurity importance) is the sum over all nodes where the feature is used to split of the impurity reduction (Equation (1) in the paper) multiplied by the proportion of training examples that fall into that node. Thus, unlike what is said in the paper, the size of the node is taken into account. As a consequence also, Gini importance also measure a feature's contribution to risk minimization.

The experimental part is interesting for comparing the different proposed implementations, as well as for comparing them against neural networks. They are lacking along several lines however.

First, I'm missing an explanation as to why quadratic loss with unbiased risk estimator is performing so poorly. In standard trees, changing Gini index for entropy does not make strong differences. Quadratic loss is also working well with the nonnegative risk estimator. I'm wondering if there is not a bug somewhere. In any case, this counterintuitive result should be discussed.

Second, some baselines are missing in the experiments that would allow to better understand how good the approach is. The comparison with neural networks is interesting but not very informative. One could change the complexity of the networks to reach better performance (or switch to CNN on the MNISTand CIFAR10 problems). I think that the comparison should include more baselines or competitors based on trees or forests to really assess how good the new impurity measures is with respect to common approaches to deal with PU learning.

For example, I would include at least a standard ET model trained on the fully labeled training set, as well as a standard ET model trained by treating all unlabeled examples as negative. This would allow to see where PU ET lies between these two extremes. I think that the comparison should actually also include some popular competitors such as those presented in [11] (as a simple improvement of the second naive baseline), [22] (as a simple adaptation of forests), and also [7]. This latter method, and the others, are excluded in the related work discussion because they are not based on the same risk estimators as PU ET in the paper. I think this is not a valid argument. One would like precisely to see whether there is any benefit in using the new risk with respect to existing approaches that can be applied with the same family of classifiers.

Two other more minor limitations of the experiments:
- Only artificial PU problems are considered that perfectly match the theoretical setting and the impact of the size of the unlabeled set is not studied.
- Only the Extra-Tree algorithm is considered. Because of its extreme randomization of the split points, the impact of the impurity measure should be very much reduced. I think it would be interesting to experiments with standard trees or RF. Even if PU ET works well, PU+DT or PU+RF could be actually worse because of some unexpected bias introduced by the new impurity measures.

Update after the authors' rebuttal:

I'm very impressed by the authors' response, which addresses almost all my complaints. I think the comparison against the baselines that I suggested proves that the approach is very strong empirically (in addition to be very well motivated theoretically). The only downside is that including all these new experiments and discussions will result in important changes with respect to the paper that we have reviewed but I'm confident that the authors will implement these changes properly.

I already recommended acceptance but I'm happy to raise my score further after this rebuttal.

---

> ### Author Response · Authors · 2022-08-05
> **Response to Reviewer 4GSs Part 1/2**
>
> Thank you for your insightful feedback! We appreciate your extensive comments.
>
> **Why uPU can cause overfitting and how nnPU solves this issue, and consequences on tree growing**
>
> *Why nnPU overfits less as compared to nnPU*: The nnPU paper [15] contains a quite technical explanation for this. We offer a hopefully intuitive enough explanation below. The uPU risk has a negative component that provides extra incentive for a classifier to try hard to fit the positive examples, thus potentially making the risk negative and leading to overfitting. nnPU corrects this behavior by forcing the sum of the negative term and the term defined on the U data to be non-negative, as the sum acts as an estimate for risk on negative examples. In addition, in line 159-164 in the paper, we provided another perspective of why nnPU avoids overfitting in the case of trees: while uPU directly minimizes the empirical risk, nnPU only minimizes an upper bound of the empirical risk, which can be considered as a regularization mechanism.
>
> *Why quadratic loss is much poorer than logistic loss for PU ET optimizing uPU risk*: Note that Eq. (11) implies that quadratic loss encourages aggressive splitting until all examples are positive as a loss of $-\infty$ can be achieved in this case, while Eq. (12) implies that logistic loss has a less aggressive splitting behavior as a loss of $-\infty$ is achieved when the total weight of positive examples exceeds the total weight of unlabeled examples - as discussed in line 199-203, once we achieve a impurity value of $-\infty$, we stop splitting. The more aggressive splitting behavior of quadratic loss leads to deeper trees that overfits more than the logistic loss in general. We consider these less intuitive aspects interesting as we derive effective ways to handle corner cases from our theory.
>
> *Comparison of uPU's and nnPU's tendency to overfit for PU ET*: We have also performed additional experiments to empirically investigate this. We report the training risks (measured as PU risk as data is PU) and testing risks (measured as PN risk as data is PN) using zero-one loss on a number of datasets. Each result is reported as mean (sd) over 5 replications. The results are shown below and we can see that the training risk is significantly smaller than the test risk in the uPU setting as compared to the nnPU setting, confirming that uPU suffers more from overfitting than nnPU.
>
>
> | &nbsp; | &nbsp; | uPU | &nbsp; | nnPU | &nbsp; |
> | -- | -- | -- |-- |-- | -- |
> | &nbsp; | &nbsp; | Quadratic | Logistic | Quadratic | Logistic |
> | 20News | Train | -0.48 (0.00) | -0.22 (0.01) | 0.00 (0.00) | 0.00 (0.00) |
> | &nbsp;| Test | 0.56 (0.00) | 0.28 (0.01) | 0.15 (0.00) | 0.18 (0.01)|
> | Mushroom | Train | -0.36 (0.00) | -0.01 (0.00) | 0.00 (0.00) | 0.00 (0.00) |
> | &nbsp;   |  Test | 0.39 (0.01) | 0.01 (0.00) | 0.00 (0.00) | 0.00 (0.00) |
> | MNIST | Train | -0.47 (0.00)  | -0.08 (0.01) | 0.00 (0.00) | 0.00 (0.00) |
> | &nbsp;| Test | 0.49 (0.00) | 0.11 (0.01) | 0.04 (0.00) | 0.06 (0.00)  |
> | CIFAR-10 | Train | -0.38 (0.00) | -0.17 (0.01) | 0.00 (0.00) | 0.00 (0.00) |
> | &nbsp;   | Test  | 0.4 (0.00) | 0.25 (0.00) | 0.16 (0.00) | 0.21 (0.00) |
> | UNSW-NB15 | Train | -0.58 (0.00) | -0.03 (0.01) | 0.02 (0.01) | 0.02 (0.01) |
> | &nbsp;    | Test  | 0.65 (0.00) | 0.13 (0.01) | 0.1 (0.01) | 0.14 (0.00) |

---

> ### Author Response · Authors · 2022-08-06
> **Response to Reviewer 4GSs Part 2/2**
>
> **Empirical comparison with other tree-based methods**
>
> We compared our PU ET algorithm with several suggested tree-based baselines to further
> study the effectiveness of our novel PU tree learning method
> * SupervisedET: ET trained on the original fully labeled dataset. This is expected to be an upper bound for other PU methods.
> * NaiveET: ET trained on the PU dataset by simply treating U data as N data.
> * PUBaggingET: PUBagging (Algorithm 1 in [22]) with Extra Tree base classifier trained on the PU dataset. We used the implementation from https://github.com/AaronWard/PU-learning-example.
>
> Hyperparameters for ET are set as in our paper. PUBagging has an additional hyperparameter: the size of bootstrap sample $K$. We used the default value of the dataset size in the reference implementation that we used.
>
> Accuracies are shown below.
>
> | Dataset   | PU-ET | SupervisedET | NaivePU-ET | PUBaggingET |
> | -- | -- | -- | -- | -- |
> | Epsilon |  57.39 (0.76) | 73.55 (00.08)   | 50.04 (0.00) | 50.04 (0.00) |
> | 20news    | 83.34 (0.22) | 85.39 (00.12)    | 43.63 (0.05) | 43.75 (0.06) |
> | Covtype   | 76.51 (0.52) | 95.90 (00.02)    | 48.71 (0.00) | 48.71 (0.0)|
> | Mushroom  | 99.70 (0.24) | 100 (00.00)      | 53.85 (0.32) | 62.76 (0.84) |
> | MNIST     | 93.60 (0.39) | 98.11 (00.05)    | 50.74 (0.00) | 50.74 (0.00)  |
> | CIFAR-10  | 79.74 (0.37) | 85.02 (00.19)    | 60.00 (0.00) | 60.0 (0.00) |
> | UNSW-NB15 | 82.24 (0.86) | 86.57 (00.05)    | 44.95 (0.01) | 44.97 (0.02)
>
> F-scores are shown below.
>
> | Dataset   | PU ET | SupervisedET | NaivePU-ET | PUBaggingET |
> | -- | -- | -- | -- | -- |
> | Epsilon   | 39.52 (1.19) | 72.96 (00.08) | 0.00 (0.00) | 0.00 (0.00) |
> | 20news    | 85.33 (0.22) | 87.40 (00.11) | 0.38 (0.19) | 0.82 (0.18) |
> | Covtype   | 75.19 (0.37) | 95.97 (00.02) | 0.05 (0.01) | 0.06 (0.01)|
> | Mushroom  | 99.71 (0.24) | 100 (00.00) | 19.68 (2.04) | 45.3 (0.84) |
> | MNIST     | 93.49 (0.42) | 98.09 (00.05) | 0.00 (0.00) | 0.00 (0.00) |
> | CIFAR-10  | 71.31 (0.43) | 80.30 (00.24) | 0.01 (0.02) | 0.02 (0.02) |
> | UNSW-NB15 | 85.65 (0.59) | 88.96 (00.04) | 0.08 (0.06) | 0.11 (0.06) |
>
> The results further support the effectiveness of our PU ET algorithm
> * Our PU ET significantly outperforms both NaivePU-ET and PUBaggingET, particularly in terms of F-scores.
> * PU ET shows strong performance even as compared to SupervisedET, though often using a small portion of the positive labels only.
>
> **Respones to other comments**
>
> * Theorem 1: Thank you for pointing out that the case for quadratic loss and Gini entropy is shown in the CART book - we have included a reference now. We have retained the statement of Theorem 1 (a) in the paper to emphasise that when one grows a decision tree in the supervised learning setting while using one of the Gini or entropy impurity measures, they are implicitly performing recursive greedy empirical risk minimization using the quadratic or logistic loss, respectively.
> * Corrections in the supplement and clarification on $v_{P',U'}^*$: We have moved corrections to the main paper and made the suggested clarification.
> * Description of PU ET with multiple thresholds per feature, and its evaluation: The algorithm is described in Algorithms 3-5 in the supplementary file, and we will clarify this by explicitly pointing out in the text that the hyperparameter $T$ is the number of thresholds to sample per feature, and $T$ could be larger than $1$ to allow sampling multiple thresholds. We have performed some additional experiments to investigate the effect of varying $T$, and found our default setting with $T=1$ works well. Please kindly refer to *(E) Tuning hyperparameter for PU ET* in our response to Reviewer jd1V for the details.
> * Feature importance formula:
> Thanks for the suggestion. For the PU case, the risk reduction importance of a feature $f$ is defined as $\sum_{n \in N_{f}} RR(f_{n}, t_{n}; P_{n}, U_{n})$, where $N_{f}$ is the set of nodes using $f$ as the splitting feature, $(f_{n}$, $t_{n})$ is the split for node $n$, and $(P_{n}, U_{n})$ is the set of PU data at $n$. This can be similarly defined for the PN case.
> * The usual definition of feature importance: Thanks for pointing us that node size is typically taken into account. We have corrected this in our paper.
> * Impact of the size of the unlabelled set, experiments with other classifiers such as DT and RF: We follow [15] using all available data as unlabeled data. We also think DT/RF variants would be interesting to explore and plan to do this in future works.

---

> > ### Comment · Reviewer_4GSs · 2022-08-07
> > **Update after the authors' rebuttal**
> >
> > I'm very impressed by the authors' response, which addresses almost all my complaints. I think the comparison against the baselines that I suggested proves that the approach is very strong empirically (in addition to be very well motivated theoretically). The only downside is that including all these new experiments and discussions will result in important changes with respect to the paper that we have reviewed but I'm confident that the authors will implement these changes properly.
> >
> > I already recommended acceptance but I'm happy to raise my score further after this rebuttal.

---

> > > ### Author Response · Authors · 2022-08-08
> > > **thank you!**
> > >
> > > Thank you for your encouraging support, and thank you again for your valuable time and insightful comments and suggestions!

---

### Author Response · Authors · 2022-08-05
**Response to all Reviewers**

We thank all the reviewers for their insightful feedback and suggestions! We have provided additional experiment results and explanations in our responses below. We believe we have addressed the reviewers' comments, and we are currently incorporating our responses in our paper. We are keen to hear the reviewers' thoughts on our responses and we will be happy to address any further questions.

---

### Meta-Review · Area_Chair_Q1Vf · 2022-08-24

**Recommendation:** Accept
**Confidence:** Certain

**Metareview:**

This paper proposes a decision tree learning approach with only positive and unlabeled examples. The reviewers are concerned that the novelty of the proposed approach is not high. However, they appreciate that the paper is clearly written, the idea of the proposed approach is reasonable, and comprehensive experiments are conducted. Therefore, I recommend accepting the paper.

**Award:**

No

---

### Decision · Program_Chairs · 2022-09-14

Accept